# HALMA: Humanlike Abstraction Learning Meets Affordance in Rapid Problem Solving

## Abstract

Humans learn compositional and causal abstraction, *i.e.*, knowledge, in response to the structure of naturalistic tasks. When presented with a problem-solving task involving some objects, toddlers would first interact with these objects to reckon what they are and what can be done with them. Leveraging these concepts, they could understand the internal structure of this task, without seeing all of the problem instances. Remarkably, they further build cognitively executable strategies to *rapidly* solve novel problems. To empower a learning agent with similar capability, we argue there shall be three levels of generalization in how an agent represents its knowledge: perceptual, conceptual, and algorithmic. In this paper, we devise the very first systematic benchmark that offers joint evaluation covering all three levels. This benchmark is centered around a novel task domain, HALMA, for visual concept development and rapid problem solving. Uniquely, HALMA has a minimum yet complete concept space, upon which we introduce a novel paradigm to rigorously diagnose and dissect learning agents' capability in understanding and generalizing complex and structural concepts. We conduct extensive experiments on reinforcement learning agents with various inductive biases and carefully report their proficiency and weakness.[1]

## 1 Introduction

Have you ever heard of *Super Halma*,[2] a fast-paced variant of *Halma*? In case you have not played *Halma* or its fast-paced variant before, we briefly introduce both of them here. *Halma* is a strategic board game, also known as *Chinese checkers*. The rules of *Halma* are minimal; it can be perspicuously explained using basic concepts of numbers and arithmetic. To win the game, one needs to transport pawns initially in one's own camp into the target camp. In each turn, a player could either *move* into an empty adjacent hole and end the play, or *jump* over an adjacent pawn, place on the opposite side of the jumped pawn, and *recursively* apply this jump rule till the end of the play. While the standard rules allow hopping over only a single adjacent occupied position at a time, *Super Halma* allows pieces to catapult over multiple adjacent occupied positions in a line when hopping; see an illustration in Fig. 1. We will use the term *Halma* to specifically refer to *Super Halma* in the remainder of the paper.

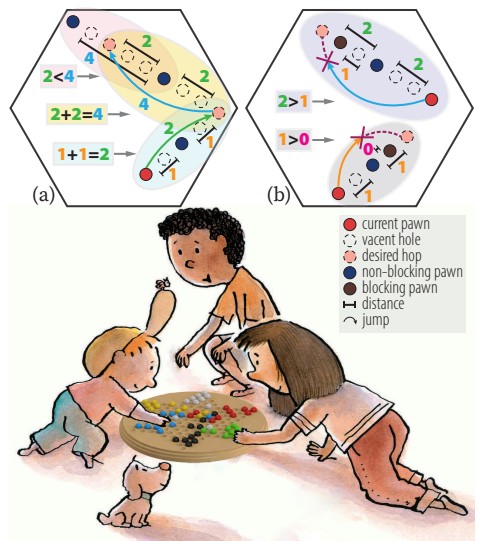

Figure 1: Illustration of the *Super Halma* playing task. By playing the game with scarce supervision, Ada should be able to learn basic concepts of numbers and arithmetic, such as concepts with both (a) valid and (b) invalid actions (jumps).

Now, imagine you are teaching your preschool cousin, Ada, to play *Halma*. Since she has not yet formed a complete notion of *natural numbers* or *arithmetic*, verbally explaining the rules to her will render in vain. Alternatively, you can play with her while providing scarce supervisions, *e.g.*, if a move is allowed; you can even reward her when she successfully moves a pawn to the target camp.

---

[1]We will make HALMA and tested agents publicly accessible upon publication.
[2]See `https://en.wikipedia.org/wiki/Chinese_checkers#Variants` for details.

By the time Ada could independently and rapidly solve unseen scenarios, we would know she has mastered the game. How many scenarios do you think Ada has to play before achieving this goal?

This *Halma* playing task is quintessential in the open-ended world; its environment is a minimal yet complete playground to test the rapid problem-solving capability of a learning agent. Under limited exposure to the underlying structure of the complex and immense concept space, we humans, by observing and interacting with entities, could form *abstract* concepts of "what it is" and "what can be done with it." The former one is dubbed *semantics* (Jackendoff, 1983) and the latter *affordance* (Gibson, 1986). These abstract concepts, once accepted as knowledge, generalize robustly over scenarios; they are considered as milestones of human evolution in abstract reasoning and general problem solving (Holyoak et al., 1996). In the case of *Halma* playing task, Ada would be able to solve unseen scenarios *within no time* if she were able to master (i) the abstract concept of natural numbers, emerged from and grounded to visual stimuli, (ii) both valid and invalid actions, and (iii) causal relations and potential outcomes risen from the grounded natural numbers and valid actions.

What is the proper machinery to learn these generalizable concepts from scarce supervisions? By *scarce supervision*, we mean the way to provide supervision is akin to how you teach Ada; one only provides sparse and indirect feedback without direct rules or dense annotations. By *generalizable concepts*, we emphasize more than the competence of memorization and interpolation; the learned representation ought to appropriately extrapolate and generalize in out-of-distribution scenarios. Such a superb generalization capability is often regarded as one of the celebrated signatures of human intelligence (Lake et al., 2015; Marcus, 2018; Lake & Baroni, 2018); it is attributed to rich *compositional* and *casual* structures in human mind (Fodor et al., 1988). Inspired by these observations, in this work, we quest for a computational framework to learn abstract concepts emerged in challenging and *interactive* problem-solving tasks, with a humanlike generalization capability: The learned abstract knowledge should be easily transferred to out-of-distribution scenarios.

The general context of interactive problem solving poses extra challenges over classic settings of concept learning; instead of merely emerging concepts, it further demands the learning agent to leverage such emerged concepts for decision-making and planning. Ada, after understanding semantics and affordance in *Halma*, can effortlessly perceive and parse novel scenarios (Zhu et al., 2020). Yet, she would still struggle in strategically playing the game as she needs to decide among multiple affordable moves. In essence, the central question is: If conceptual knowledge can generalize as such, what meta-benefits does it offer on solving unseen problems (Schmidhuber et al., 1996)? The classic decision-making account of these meta-benefits would be: Leveraging knowledge, we can develop cognitively executable *strategies* with high planning (Sanner, 2008) and exploration efficiency (Kaelbling et al., 1998); these strategies facilitate us to solve problems rapidly in unseen scenarios. They are what we call the *algorithms* or *heuristics* of this task. Taking a step further, Wang et al. (2018); Guez et al. (2019) hypothesize that modern reinforcement learning agents, incentivized by these meta-benefits, have already discovered such algorithms. However, to date, their argument is still speculative since these agents have not been evaluated in tasks with rich internal structures yet limited exposure (Lake et al., 2017; Kansky et al., 2017). A diagnosis benchmark for generalization capability is thus in demand to bridge communities of concept development and decision-making.

The main contribution of this paper is a *Halma*-inspired competence benchmark: Humanlike Abstraction Learning Meets Affordance (HALMA). We rigorously devise HALMA with three levels of generalization in visual concept development and rapid problem solving; see details in Section 2. HALMA is unique in its *minimum yet complete* concept spaces, a miniature of compositional and causal structures in human knowledge. It *dynamically* generates test problems to informatively evaluate learning agents' capability in out-of-distribution scenarios *under limited exposure*. We conduct extensive experiments with reinforcement learning agents to benchmark proficiency and weakness.

## 2 THREE LEVELS OF GENERALIZATION

Our motivations might seem, *prima facie*, bold. To convince readers and support our optimism, we summarize some recent progress in this section. In particular, we provide a taxonomy of three levels of generalization on a competency basis. Indeed, generalization is a multifaceted phenomenon. Previous evaluations for generalization were predominantly defined in a statistical sense, following the classical paradigm of train-evaluation-test random split (Cobbe et al., 2019) while ignoring internal structures. However, we argue this classical paradigm should not be the only objective approach wherein agents can or should generalize beyond their experience (Barrett et al., 2018), especially if our goal is to construct humanlike general-purpose problem-solving agents (Lake et al., 2017).

**Perceptual Generalization** Perceptual generalization characterizes agents' capability to represent unseen perceptual signals, *e.g.*, *appearance* or *geometry* in vision. In his seminal book, *Vision*, Marr (1982) describes the process of vision as constructing a set of representations, parsing visual sensory data into descriptions. Such descriptions provide *conceptual primitives* (Carey, 2009) for agents' understanding of the environment, boosting the efficacy of downstream cognitive activities (*e.g.*, memory, learning, and reasoning). Learning an object-oriented representation of independent generative factors without supervision is thus believed to be a crucial precursor for the development of humanlike artificial intelligence. Although unsupervised disentanglement and segmentation (Eslami et al., 2016; Higgins et al., 2017) resurged years ago, it is only till Locatello et al. (2019) did we realize the importance of evaluation on their generalization. More recently, Burgess et al. (2019), Greff et al. (2019), and Lin et al. (2020) evaluate their disentanglement/segmentation models outside of training regimes, especially on unseen combinations of visual attributes and numbers of objects.

Although a hypothetically perfect *semantic* description can truthfully represent the primitive concept of "what it is," it could only contribute partially to achieving the understanding of "what can be done with it" (Montesano et al., 2008; Zhu et al., 2015). Humanlike agents should equip with such task-oriented abstraction, *affordance*, supported by compelling evidences in the field of developmental psychology; for instance, 18 to 24-month-old infants can distinguish *bootstrapped concepts* (Quine, 1960), such as "a walkable step is not a cliff" (Kretch & Adolph, 2013). At a computational level, given a task specified by a Markov decision process, irrelevant features should be *abstracted out* (Li et al., 2006; Ferns et al., 2011; Khetarpal et al., 2020). Representation learned in this way bootstraps conceptual content. Recently, disentanglement as such has demonstrated efficacy (Gelada et al., 2019; Wayne et al., 2018) and elementary perceptual generalizability (Zhang et al., 2020).

**Conceptual Generalization** While perceptual generalization closely interweaves with vision and control, conceptual generalization resides completely in cognition, assuming the readiness of all primitive concepts and some bootstrapped ones. The central challenge in conceptual generalization[3] is: How well can an agent perform in unseen scenarios given *limited exposure* to the underlying *configurations* (Grenander, 1993)? It is connected with the Language of Thought Hypothesis (Fodor et al., 1988; Goodman et al., 2008): The productivity, systematicity, and inferential coherence in languages characterize compositional and causal generalization of concepts (Lake et al., 2015).

How to learn representations with conceptual generalization is still an open question, drawing increasing attention in our community. With a synthetic translation task, Lake & Baroni (2018) reveal the incompetence of general purpose recurrent models (Elman, 1990; Hochreiter & Schmidhuber, 1997; Chung et al., 2014) in generalizing to (i) unseen primitives, (ii) unseen compositions, and (iii) longer sequences than training data. Similar incompetence of relational inductive biases (Battaglia et al., 2018) on hard compositional extrapolation has also been exemplified in abstract visual reasoning (Barrett et al., 2018). Notably, there is also a line of research on *emerging* these linguistic structures from bootstrapped communication (Lazaridou et al., 2018; Mordatch & Abbeel, 2018).

**Algorithmic Generalization** Agents' understanding of the structured environment should be reflected in their performance in solving novel problem instances; they ought to build strategies upon the developed concepts, resembling *cognitive control* in human mind (Rougier et al., 2005; Botvinick & Cohen, 2014). We use the term algorithmic generalization to describe such flexibility. Specifically, for a problem domain where the internal structure contains an optimal exploration strategy, algorithmic generalization requires agents to discover this optimal strategy to explore efficiently in *new* problem instances. For example, in the domain of dependent bandit problems designed by Wang et al. (2016), there is one arm whose return leaks the index of the optimal arm. Given a new problem, agents who discovered the algorithm of this domain would first try the leaky arm and then go straight to the optimal arm. Furthermore, as an acid test, algorithmic generalization also measures the agent's ability in long-term planning in unseen problem configurations, after acquiring adequate information. Evaluation as such has been discussed by Tamar et al. (2016) and Guez et al. (2019).

Problem domains discussed above, however, still lack rich concept spaces, nor do they test agents' perceptual generalization, omitting the interaction among the three levels introduced in this paper. Essentially, they are still far-off from the famous Atari game, Frostbite, which is argued to be a testbed for humanlike problem solving (Lake et al., 2017). In this work, we introduce a new problem domain to facilitate joint efforts towards representations with these three levels of generalization.

---

[3]Conventionally, it is dubbed *combinatorial* generalization or *systematic* generalization. We use the term *conceptual* to highlight its functional signature.

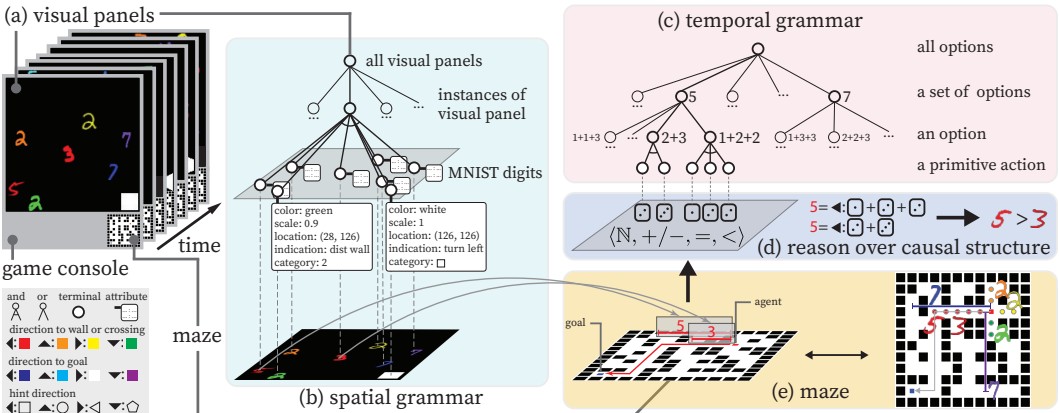

Figure 2: Illustration of the HALMA basics (see Section 3.1), problem generation, and concept space (see Section 3.2). (a) Given a visual panel with various colored MNIST digits and a hint, an autonomous agent is tasked to reach the goal in a maze. The concept space guides the generation of the visual panels; it consists of (b) spatial grammar, (c) temporal grammar, and (d) causal structure. (e) The semantics and affordance of the colored MNIST digits are augmented on the corresponding maze; the maze is not shown to the agent.

# 3 HUMANLIKE ABSTRACTION LEARNING MEETS AFFORDANCE (HALMA)

## 3.1 HALMA BASICS

The setup of HALMA is minimal and interpretable. Instead of replicating the entire game of *Halma*, we only preserve the most essential ingredients: The learning agent is cast as one pawn, navigating around the "magical" *Halma* landscape by itself. To simplify the environment without lost of generality, we build a maze in a grid-world for each *scenario* (or *problem* henceforth), resembling a *cognitive map* of the agent. Distinct from vanilla grid-world maze games, HALMA is novel in terms of our design of its observation space and action space. The agent perceives neither the global map nor any local patch of the global map; instead, it is shown with a visual panel of various numbers of MNIST digits in various color, randomly scaled and placed; see Fig. 2 (a). These colored digits indicate the *semantics* of (i) the distance till a wall towards each direction, (ii) the distance till the nearest crossing or T-junction towards each direction, and (iii) the distance and direction to the goal; the visual panel only displays non-zero distances. For example, in Fig. 2 (a) (e), **5** indicates the wall to the left is 5-grid away, and **3** indicates the nearest crossing is is 3-grid away to the left; the visual color of red refers to the semantics of "left." The agent will also be hinted with a symbol from the set $\{\bigcirc, \triangle, \square, \diamondsuit\}$ at any crossing for the correct direction; see an example of $\square$ in Fig. 2 (a). When making a decision, the agent needs to first select a direction and then select either a primitive action or an option composed by a sequence of primitive actions (Sutton et al., 1999) with maximum length `max_opt_len`. The direction set is $\{\blacktriangle, \blacktriangledown, \blacktriangleleft, \blacktriangleright\}$. The primitive action set, in terms of the number of moves, is $\{\boxdot, \boxdot, \boxdot, \boxdot\}$; this design of primitive numbers with a maximum of three aligns with the doctrine of core knowledge in developmental psychology (Feigenson & Carey, 2003; Dehaene, 2011). If an option is selected, consecutive hops as in *Halma* are simulated; all observations from intermediate states will be skipped, and only the observation of the final state is provided. A move would fail if a wall stops the agent, leaving the agent's position unchanged; failure moves bring penalties to the agent. The agent would receive a positive reward when reaching the goal. Such a design encourages the agent to comprehend which MNIST digit *affords* it to take which moves.

Essentially, HALMA is a 2D contextual navigation game, sharing the same spirit with those in Mirowski et al. (2017) and Ritter et al. (2018). However, *contexts* in these prior works are elusive and conceptually meaningless. As such, they only evaluate generalization at either the visuomotor or algorithmic level. In stark contrast, HALMA is unique, possessing a rich, crisp, and challenging configuration space of problems, semantics, and affordance; see details in the next subsection.

## 3.2 PROBLEM GENERATION AND CONCEPT SPACE

Generating a HALMA problem consists of two sub-procedures: (i) generating a grid-world maze problem with *valid* optimal paths, and (ii) producing a set of visual panels, based on an explicit spatial grammar of the *concept space*, that uniquely represent observations in the maze.

Generating a grid-world maze problem is intricate since HALMA is a partially observable game. A randomly generated maze may perplex the agent with ambiguous observations that hinders the agent's formation of a coherent strategy; see Appendix A for an example. To alleviate this issue, instead of first generating a complete maze and then producing optimal paths, our solution is to reverse this process by first generating *valid* optimal paths and then adding deceptive branches to construct a grid-world maze. Formally, a path is said to be *invalid* if an agent who possesses an oracle understanding of the concept space fails to make the oracle decision; such a definition of *validity* is deeply rooted in the concept space that the agent is required to learn. We refer the readers to check Appendix A for an example of *invalid* optimal path, an example of a successfully generated maze with a *valid* optimal path, an example sequence, and additional implementation details.

Producing visual panels heavily relies on the concept space. The concept space of HALMA consists of an explicit *spatial grammar* for visual panels, an implicit *temporal grammar* for actions and options, and an underlying *causal structure* that specifies the intersection of spatial and temporal grammar. For simplicity, we only introduce them verbally here; see an illustration in Fig. 2 and their formal definitions in Appendix B. Intuitively, the spatial grammar produces all possible descriptions of visual panels, spanning all configurations of *semantics* introduced in Section 3.1. To generate a visual panel for a given state, we first sample an MNIST digit for each entry of its description and then sample a random scale and position. The sampled MNIST digit is then colored on the basis of its semantics, *i.e.*, directions to a wall, a crossing, or a goal; see Fig. 2 (b) and the legend. The temporal grammar produces all possible moves, either a single primitive action or a composed option, regardless of the visual stimuli. For instance, a non-terminal node ◀ : 5 can be parsed into options opt, such as ◀ : ⚀ + ⚀ + ⚀ and ◀ : ⚀ + ⚁; see Fig. 2 (c). Despite of their distinction in terms of how an option is decomposed into primitive actions, these options are equivalent in their causal effects. Specifically, these causal effects bind visual MNIST digits with digital actions based on one of the simplest mathematical structures in human cognition (Flavell, 1963): $\langle \mathbb{N}, +/-, =, < \rangle$; namely, *natural numbers* $\mathbb{N}$, *operations* $+/-$, and *relations* $=, <$ over $\mathbb{N}$. For example (see also Fig. 2 (d)), a learning agent is expected to understand relations between **5** and **3** via

- $\langle S, < \rangle$: the set of semantic *generators*[4] with an *order* over it, *e.g.*, **3** < **5**;
- $\langle A, +/-, = \rangle$: the set of affordance generators with operations and equality, *e.g.*, **5** = ◀ : ⚀ + ⚀ + ⚀ = ◀ : ⚀ + ⚁ = ...;
- $\langle A, +/-, < \rangle$: the set of affordance generators with operations and inequality, *e.g.*, ◀ : ⚀ + ⚀ < **5**, **5** < ◀ : ⚀ + ⚀ + ⚁;
- $\langle C, +/-, = \rangle$: the set of causal generators with operations and equality, *e.g.*, **5** = **3** + ◀ : ⚀.

## 3.3 TASK FORMULATION AND EVALUATION

We expect agents who developed the concept space to leverage this knowledge and rapidly solve new problems in HALMA. To this end, we formulate this rapid problem-solving task with an objective to *maximize the agent's rewards accumulated over a few trials in a **novel** problem instance*:

$$\mathbb{E}_\zeta \Big[ \sum_{i=0}^{N} \gamma^{\sum_{j=0}^{i-1} \text{len}(\tau_j)} \sum_{t=0}^{\text{len}(\tau_i)-1} \gamma^t R(s_{\tau_i,t}, a_{\tau_i,t}) \Big]. \tag{1}$$

Specifically, an agent's experience in each problem instance is dubbed an *episode* $\zeta$ (Wang et al., 2016), which terminates when a maximum number of *steps* $L$ is reached or a maximum number of *trials* $N$ have been accomplished. A *trial* $\tau$ proceeds with actions $a_{\tau,t}$, spanning multiple *steps* $t$; it starts from an initial state $s_0$ and terminates when the agent reaches the goal $s_g$ (thus accomplished), or when it consumes the maximum number of steps $H$ (thus failed). The agent is respawned to the initial state when a trial terminates. It is awarded $R(s_g, \cdot)$ if the trial is accomplished. The cumulative reward in one episode is the sum of temporally $\gamma$ decayed accomplishments. When one episode terminates, the agent is presented with the next problem.

Under this task formulation, learning agents should be evaluated against oracle solutions, analogous to ground-truth annotations in supervised learning; recall that the oracle agent has complete understanding of the concept space and the problem domain. Since HALMA is a *partially observable* domain, its oracle behavior consists of two aspects: optimal exploration and optimal planning. As introduced in Section 3.2, problems are generated by adding deceptive branches to optimal paths. Hence, the optimal exploration strategy is to stop at each crossing to obtain the hint from the visual

---

[4]For the sake of formalism, we adopt the terminology from General Pattern Theory (Grenander, 1993), wherein the term *generator* refers to basic units in a *configuration space*. Intuitively, an object file (Kahneman et al., 1992), is a semantic generator. It is also a generator for configuration spaces of affordance and causality, for which actions/options are also generators. We refer the readers to Appendix B for detailed formal definitions.

panel. Intuitively, the agent should understand "when two digits with the same color are exhibited in the visual panel, the *lesser* one indicates the crossing, and I should stop there for hint" based on the concept of $\langle \text{S}, < \rangle \cup \langle \text{A}, +/-, < \rangle$. An oracle agent would sacrifice the first trial to explore; note that the cost is still low as it would explore along the optimal path with the guidance of hints, avoiding all deceptive branches. Afterwards, the oracle agent should retrieve its experience and merges consecutive moves towards the same direction to form the optimal plan. Take the maze example shown in Fig. 2 (e); during exploration, the agent sees a 5 and a 3 in the visual panel and takes an option ◀ : ⊡ + ⊡ to obtain a hint □, which guides it to keep moving left ◀ : ⊡ until the wall. Then in the second trial, the agent should exploit $\langle \text{A}, +/-, = \rangle \cup \langle \text{C}, +/-, = \rangle$ via ◀ : ⊡ + ⊡ + ⊡. With this oracle agent, we can have evaluation metrics normalized across different problems. Instead of directly calculating the ratio of Eq. (1) between proposed agents and the oracle agent, which involves strong non-linearity, we carefully decompose it into three metrics with more intuitive measures:

- Ratio of invalid moves $\rho_a = \mathbb{E}_\zeta \left[ \frac{\#\text{invalid moves}}{\sum_i \text{len}(\tau_i)} \right]$ for semantics and affordance understanding;
- Success rate of goal reaching $\rho_g = \mathbb{E}_\zeta \left[ \frac{1}{N} \sum_i \delta(s_{\tau_i, -1} = s_g) \right]$ for leveraging concepts to explore;
- Efficiency in exploration and planning $\rho_p = \mathbb{E}_\zeta \left[ \frac{1}{N} \sum_i \frac{\text{len}(\tau^\star)}{\text{len}(\tau_i)} \right]$ for algorithmic understanding.

### 3.4 GENERALIZATION TEST

One of our key contributions in HALMA is a novel paradigm to test agents' capability in all three levels of generalization, which extends the classical paradigm of statistical learning. Our training set consists of 100 mazes[5] along with their visual panels; we summarize the statistics of these visual panels in Appendix C to show that the generated dataset is balanced, yielding fair distributions of crucial statistics. Different from the classic paradigm, the evaluation of agent's performance in HALMA would emphasize on the *explicit extrapolation* test, which should be conducted in the *held-out* compositional and relational configurations; such design echoes recent trend in evaluating agent's generalization capability (Burgess et al., 2019; Lake & Baroni, 2018; Zambaldi et al., 2019). Compared to these prior domains, HALMA is unique as it is a partially observable and interactive problem-solving task, wherein an agent is tasked to *autonomously* learn the immense concept space and form the abstract knowledge. Hence, simply holding off a *pre-selected*, *fixed* subset of conceptual configurations would impose severe restrictions on problem generators. For instance, if we would like to allow agents to see a 4, they must be able to see a 3 by simply moving ◀ : ⊡ from where they see 4. In other words, if we managed to strictly withhold 3 from agents, they would not see any red digits larger than 3 in this *interactive* problem solving task. Therefore, an *ex post* evaluation protocol that *dynamically* generates tests is more desirable.

In this paper, we propose an ingenious solution: Instead of *aimlessly* generating a large test set of *random* cases, we devise an algorithm to *proactively* generate *tailored* tests in accord to what the agent might have learned; this design would produce a definitive and much more informative evaluation of agent's competence. The intuition is simple: When a teacher finds a student consistently make right decisions during training, wherein the student only needs to understand 3 < 5 and 4 = 2 + ◀ : ⊡, the teacher may quiz the student on 3 vs 5 and 2 vs 4. To implement this protocol in HALMA, we first store agents' experience during training as their external memory MEM. We then construct a representation to emulate agents' *knowledge bases* (KB) for $\langle \text{S}, < \rangle$ and $\langle \text{A}, +/-, = \rangle \cup \langle \text{C}, +/-, = \rangle$: $\text{KB}_\text{S}$ tracks the agent's understood configurations on semantics, and $\text{KB}_{\text{A} \vee \text{C}}$ tracks the agent's understood configurations on affordance and causality. Here, we assume that (i) valid decisions[6] in experience were made upon understanding *inequality* configurations, and (ii) agents understand configurations involving *equality* and *operations* in experienced transitions. With these KBs, we *dynamically* generate test problems with novel configurations, wherein agents should likewise act appropriately if they understood not only seen configurations but their underlying concepts; see details of constructing KBs and generating test problems in Appendix D.

Tests in HALMA are on the competence basis: Conceptual generalization is built upon perceptual generalization, with the algorithmic generalization resides on top. Tests for perceptual generalization are backed by the *spatial grammar*, including unseen MNIST images and unseen compositions of visual attributes, *i.e.*, shape and color. Tests for conceptual generalization are based on *the concept of* $\langle \mathbb{N}, +/-, =, < \rangle$, consisting of novel *equality* and *inequality* configurations. Results of these two

---

[5]This design reflects our thesis argument, *i.e.*, agents shall generalize their understanding from limited exposure to the concept space. An ablation study on the volume of training set can be found in Appendix G.1.

[6]Note that some decisions may come from random exploration. We introduce a threshold on the visitation count to filter them out.

tests are manifested in algorithmic generalization. Specifically, agents could only pass all of these tests by making right *exploration* decisions based on relations of novel digit pairs $\langle d_1, d_2 | \texttt{type} \rangle$, where $\texttt{type}$ refers to various directions. Inappropriate exploration may cause agent to miss hints at crossings or to be trapped in dead-ends, resulting in failures of the tests. Moreover, these novel digit pairs also test the agents' understanding of the *temporal grammar*, requiring agents to make proper *exploitation* decisions by merging novel consecutive actions/options into a *greater* option.

Since conceptual generalization connects the other two, all three levels of generalization are covered when test problems are dynamically generated with novel configurations in $\langle \mathbb{N}, +/-, =, < \rangle$. Recall that the generation mechanism of a problem is to first generate an unseen configuration of optimal path and then add deceptive branches; the latter is pivotal for a test problem since it involves generating novel digit pairs $\langle d_1, d_2 | \texttt{type} \rangle$. By design, the lesser digit within a pair should indicate the distance to the nearest crossing, and the greater the distance to the wall. Hence, agents could be tested by these novel digit pairs, queried based on the agent's KBs. We categorize the problems into:

- Semantic Test (**ST**): $\texttt{KB}_{\texttt{ST}} = (\langle d_1, d_2 | \texttt{type} \rangle \notin \texttt{KB}_{\texttt{S}}) \wedge (\exists_x \langle d_1, d_2 | x \rangle \in \texttt{KB}_{\texttt{S}})$, *i.e.*, testing visual panels differentiated from $\texttt{KB}_{\texttt{S}}$ in terms of color, shape, or other MNIST digits.
- Affordance Test (**AfT**): $\texttt{KB}_{\texttt{AfT}} = (\forall_x \langle d_1, d_2 | x \rangle \notin \texttt{KB}_{\texttt{S}}) \wedge ((\exists \langle d_1, d_2 | x \rangle \in \texttt{KB}_{\texttt{A} \vee \texttt{C}}) \vee (d_1 = \texttt{opt}_1 \in \texttt{KB}_{\texttt{A} \vee \texttt{C}} \wedge d_2 = \texttt{opt}_2 \in \texttt{KB}_{\texttt{A} \vee \texttt{C}}))$, *i.e.*, testing inequalities inferred from equalities in $\texttt{KB}_{\texttt{A} \vee \texttt{C}}$. $\texttt{opt}$ denotes actions or options.
- Analogy Test (**AnT**): $\texttt{KB}_{\texttt{AnT}} = (\forall_x \langle d_1, d_3 | x \rangle \notin \texttt{KB}_{\texttt{ST} \vee \texttt{AfT}}) \wedge (\exists \{ \langle d_1, d_2 | x \rangle, \langle d_2, d_3 | x \rangle \} \subset \texttt{KB}_{\texttt{ST} \vee \texttt{AfT}}) \wedge (\exists \{ \langle d_1', d_2' | x \rangle, \langle d_2', d_3' | x \rangle, \langle d_1', d_3' | x \rangle \} \subset \texttt{KB}_{\texttt{ST} \vee \texttt{AfT}})$, *i.e.*, testing inequalities inferred from the *transitivity* of $<$. $\texttt{KB}_{\texttt{ST} \vee \texttt{AfT}} = \texttt{KB}_{\texttt{ST}} \cup \texttt{KB}_{\texttt{AfT}}$.

Specific examples of these tests can be found in Table 1. See Appendix D for detailed explanation.

## 4 MODELS AND EXPERIMENTS

The motivating questions of our experiments are: (i) Do model-free agents, exploiting generic inductive biases, develop concepts that generalize in a way, akin to human knowledge? (ii) If there are indeed certain meta-benefits induced by these architectural priors towards problem solving, are they achievable with only limited exposure to the concept space? As it is logistically challenging to experiment with all existing models, a representative subset is culled for benchmark: model-free reinforcement learning agents (Wang et al., 2016; Zambaldi et al., 2019) with gated memory mechanism (Hochreiter & Schmidhuber, 1997), self-attention mechanism (Vaswani et al., 2017), or both. Notably, Wang et al. (2016) argued that when an RNN agent is fed with previous actions and rewards, its LSTM module would emulate an inner reinforcement learning algorithm; the agent is thus learning to reinforcement learn. They demonstrated that the learned exploration strategy is more efficient than a near-optimal model-free exploration algorithm. Zambaldi et al. (2019) argued that by exploiting stacked attention modules, Transformer agents can conduct iterated reasoning with seen relational units and generalize to unseen scenarios. By our evaluation protocol, however, these prior models did not demonstrate conclusive evidence to support all three levels of generalization proposed in this paper; hence, the precise level of generalization is obscure. Crucially, neither of them evaluated the learned agents *under limited exposure* to a *complex concept space* as in HALMA.

Table 1 shows the full list of agents used in our experiments; see Appendix E for implementation details. All agents are trained with an off-the-shelf reinforcement learning method, TD3 (Fujimoto et al., 2018). All agents' policies converged at the end of training.

To decouple the evaluation of conceptual generalization from perceptual generalization, we first conduct experiments with symbolic one-hot observations, which can be regarded as the ground-truth representation of perception; see details of this observation space in Appendix F.1. All agents show relatively high invalid action ratio $\rho_a$ in tests of random split, indicating their understanding of affordance is brittle even with the ground-truth semantics. Under this precondition, we find that all agents can still perform relatively well in terms of goal-reaching $\rho_g$ and efficiency $\rho_p$ in random splits. However, when transferred to our generalization tests, MLP agents exhibits a significant degradation. Agents with LSTM modules, on the contrary, can somehow maintain or even surpass their $\rho_g$ and $\rho_p$ in training problems. One possible explanation to their high $\rho_g$ is: With a memory mechanism, they learn to recover from dead-ends even if they missed the hints at crossings. Even though they also have higher $\rho_p$ than MLP agents, consistent with the findings reported by Wang et al. (2016), this measure is still disconcertingly low. Such low performance implies that agents do not understand the concept space well, especially in terms of the temporal grammar. Transformer agents do perform better than MLP agents in generalization tests, but not as good as LSTM agents. In particular, even though Zambaldi et al. (2019) argued that Transformer agents as such may learn

Table 1: Examples and results of generalization tests (- indicates no problem is dynamically generated)

| Test Type & Examples | | % | SYMBOLIC (max_opt_len=5) | | | | VISUAL (max_opt_len=1) | | |
|---|---|---|---|---|---|---|---|---|---|
| | | | MLP | LSTM | TRAN | TRAN+LSTM | CNN+MLP | CNN+TRAN | SPACE |
| T | Training problems | $\rho_a\downarrow$ | 5.22±4.11 | 12.12±2.14 | 14.57±6.77 | 13.05±3.09 | 14.39±7.22 | 10.29±2.61 | 16.45±2.65 |
| | | $\rho_g\uparrow$ | 99.23±0.63 | 57.22±3.07 | 93.85±1.26 | 72.33±5.79 | 75.76±4.77 | 58.33±4.19 | 16.33±0.94 |
| | | $\rho_p\uparrow$ | 71.67±1.73 | 50.91±3.54 | 67.89±0.63 | 63.97±5.84 | 63.77±2.68 | 35.31±3.00 | 12.02±1.17 |
| RT | Random split | $\rho_a\downarrow$ | 37.02±1.52 | 23.91±2.10 | 34.85±4.45 | 37.69±2.90 | 86.70±2.30 | 56.91±7.92 | 58.38±1.20 |
| | | $\rho_g\uparrow$ | 51.00±2.21 | 57.78±3.49 | 82.82±0.96 | 54.00±2.94 | 7.58±0.43 | 14.00±4.24 | 3.67±0.47 |
| | | $\rho_p\uparrow$ | 54.91±2.85 | 45.15±1.46 | 58.07±1.01 | 40.13±2.52 | 5.09±1.17 | 8.33±1.96 | 2.66±0.19 |
| ST | ⟨3,5,1,7⟩ ∈ MEM, test ⟨3,5,4,2⟩ ∉ MEM. | $\rho_g\uparrow$ | 55.00±7.07 | 50.00±8.16 | 41.67±8.50 | 66.67±13.12 | 0.00±0.00 | 0.00±0.00 | 0.00±0.00 |
| | | $\rho_p\uparrow$ | 19.90±2.18 | 24.02±7.20 | 16.34±3.90 | 35.74±5.85 | 0.00±0.00 | 0.00±0.00 | 0.00±0.00 |
| | 3 < 5 ∈ KB$_S$, test ⟨3,5⟩ ∉ KB$_S$. | $\rho_g\uparrow$ | 25.00±8.16 | 63.33±6.24 | 43.33±6.23 | 78.33±2.36 | 0.00±0.00 | 0.00±0.00 | 0.00±0.00 |
| | | $\rho_p\uparrow$ | 7.37±2.33 | 26.31±2.34 | 12.22±1.83 | 34.79±4.25 | 0.00±0.00 | 0.00±0.00 | 0.00±0.00 |
| Aff | 5 = 3 + ◄ : □ ∈ KB$_{AvC}$, test ⟨3,5⟩ ∉ KB$_S$. | $\rho_g\uparrow$ | 41.67±2.36 | 60.00±10.80 | 36.67±8.50 | 58.33±10.27 | 0.00±0.00 | 0.00±0.00 | 0.00±0.00 |
| | | $\rho_p\uparrow$ | 15.10±0.35 | 28.91±7.62 | 14.01±3.75 | 27.11±2.12 | 0.00±0.00 | 0.00±0.00 | 0.00±0.00 |
| | {5 = ▲ : □ + □, 3 = ▲ : □} ⊂ KB$_{AvC}$, test ⟨3,5⟩ ∉ KB$_S$. | $\rho_g\uparrow$ | 31.67±8.50 | 45.00±10.80 | 43.33±6.24 | 71.67±6.24 | 0.00±0.00 | 0.00±0.00 | 0.00±0.00 |
| | | $\rho_p\uparrow$ | 11.68±3.34 | 17.15±5.82 | 17.86±3.02 | 35.40±3.71 | 0.00±0.00 | 0.00±0.00 | 0.00±0.00 |
| | 5 = 3 + ▲ : □ ∈ KB$_{AvC}$, test ⟨3,5⟩ ∉ KB$_S$. | $\rho_g\uparrow$ | 6.67±2.36 | 100.00±0.00 | 25.00±0.00 | - | 0.00±0.00 | 0.00±0.00 | 0.00±0.00 |
| | | $\rho_p\uparrow$ | 1.48±0.52 | 51.86±0.18 | 5.83±0.24 | - | 0.00±0.00 | 0.00±0.00 | 0.00±0.00 |
| | {5 = ▶ : □ + □, 3 = ▼ : □} ⊂ KB$_{AvC}$, test ⟨3,5⟩ ∉ KB$_S$. | $\rho_g\uparrow$ | 0.00±0.00 | 86.67±9.43 | 50.00±0.00 | - | 0.00±0.00 | 0.00±0.00 | 0.00±0.00 |
| | | $\rho_p\uparrow$ | 0.00±0.00 | 29.89±2.18 | 10.00±0.00 | - | 0.00±0.00 | 0.00±0.00 | 0.00±0.00 |
| AnT | {3 < 4, 4 < 5, 3 < 5, 6 < 7, 7 < 8} ⊂ KB$_{STvAfT}$, test ⟨6,8⟩ ∉ KB$_{STvAfT}$. | $\rho_g\uparrow$ | 35.00±7.07 | 48.33±4.71 | 41.67±2.36 | 41.67±13.12 | 0.00±0.00 | - | 0.00±0.00 |
| | | $\rho_p\uparrow$ | 12.19±1.84 | 21.84±0.53 | 14.45±1.66 | 22.03±6.64 | 0.00±0.00 | - | 0.00±0.00 |

to plan, their lower $\rho_p$ in HALMA task implies the opposite, at least under partial observation without a memory mechanism. Combining the benefits from the attention and the memory mechanisms, TRAN+LSTM agents outperform others in almost all generalization tests on both $\rho_g$ and $\rho_p$. Another interesting phenomenon is: By removing the constraint of *limited exposure* (*e.g.*, we increase the training volume to 10×), all agents, no matter what inductive biases are encoded, achieve around 80% measured by $\rho_g$, and those with LSTM modules have $\rho_p$ at around 45%; see details in Appendix G.1. Since no state-of-the-art agents could pass the test on $\rho_p$, we summarize the results of symbolic experiments as: In the spectrum of model-based vs model-free, emerged strategies still reside on the model-free side of the oracle agent. Significant efforts are needed to devise agents capable of humanlike conceptual and algorithmic generalization.

Under visual observation, however, all agents fail the generalization test when simply connected with a convolutional module, even in the easiest setup (max_opt_len=1). Assuming CNNs do not offer sufficient priors to induce an object-oriented, independently disentangled representation, we pretrain a state-of-the-art multi-object segmentation and disentanglement model, SPACE (Lin et al., 2020), with all visual panels in the training set. The converged model exhibits remarkable generalization in reconstruction, segmentation, and detection, consistent with the results reported by Lin et al. (2020); see details in Appendix F.3. One would expect that, by connecting the encoder of this powerful pretrained visual module with an RL agent using a Transformer module for the object-oriented encoding, the model would have a superb performance. Counter-intuitively, our results show that SPACE agents perform worse than CNN+TRAN agents even under random split. A further investigation reveals that the latent space of object slots fails to disentangle shapes or colors (*e.g.*, 3 vs 5), even though they can be substantially distinguished and reconstructed by the strongly nonlinear decoder. This explanation also accounts for SPACE agents' high invalid action ratio in test problems ($\rho_a = 58.38 \pm 1.20$). In principle, they misunderstand affordance because they fail to recognize "what it is" in the first place. More details on this SPACE experiment can be found in Appendix F.3. Taking together, we argue that HALMA does extend the evaluation paradigm of perceptual generalization, posing new challenges to the community of unsupervised disentanglement.

## 5 RELATED WORK

Recently, there emerges a burst-out of benchmarks for diagnosing a set of clearly defined competencies of AI systems, which we draw inspiration from and sincerely honor. In a word, HALMA differentiates from all of them in its holistic evaluation towards all three levels of generalization.

Readers may be curious about the relation between HALMA and conventional navigation tasks such as Mirowski et al. (2017). We hope we have made it clear the difference between HALMA and them in Section 3.1 of main text: In these navigation tasks, there is only one maze, and new problem instances are simply new combinations of initial and goal states. Hence, rapid problem solving only requires agents to memorize the whole maze, whereas in HALMA the only shared structure between problem instances is the concept space. Going beyond memorization, HALMA requires two extra cognitive abilities—understanding and reasoning. We also notice that in another embodied navigation task, the Habitat challenge (Savva et al., 2019), agents are indeed evaluated in completely unseen environments, under the protocol of which Wijmans et al. (2020) has achieved close-to-optimal

performance with large-scale training. However, without a clearly specified concept space, the evaluation in Habitat is akin to the Random Split in HALMA under the setup of `max_opt_len=1`. The reason why we emphasize `max_opt_len` is that the very idea of *affordance* is only interesting if the action/option space is large enough and highly structured. Otherwise, when `max_opt_len=1`, agents with memory or attention do generalize well in both Random Split and our Dynamic Test; see detailed results in Appendix G.2. Perhaps the notion of *affordance* seems a bit abstract in HALMA and can be more intuitive in visual semantic navigation and control (Yang et al., 2019; Chaplot et al., 2020). We hope our work can inspire the future development of benchmarks for these topics.

Compositional Language and Elementary Visual Reasoning (CLEVR) (Johnson et al., 2017) is one of the earliest datasets that diagnose models' visual reasoning abilities. High-level reasoning skills required in CLEVR include counting, comparing, logical inference, and memory. The same set of skills are also required in HALMA, but without the guidance of language. Accounting for a similar purpose, Bahdanau et al. (2019) propose a minimalist alternative, Spatial Queries On Object Pairs (SQOOP). While relations in SQOOP are only spatial, benchmarks inspired by Raven's Progressive Matrices (RPM) are proposed towards abstract visual reasoning (Barrett et al., 2018; Zhang et al., 2019), in which the capacity of sequential decision making is not required. In sum, all prior works listed in this paragraph are discriminative tasks. Different from them, the generative nature of interactive problem solving in HALMA is akin to human exploration in the open-ended world.

As for planning and reinforcement learning, Box-World and StarCraft II minigames (Vinyals et al., 2017) in Zambaldi et al. (2019) are tasks that also require relational concept learning; the concepts within, however, are mostly spatial. In contrast, the concept space in HALMA is abstract and complex. The mapping from the visual space to the semantic space is non-trivial to learn, which requires agents' understanding of the temporal grammar and the causal structure. Moreover, HALMA is a partially observable domain that requires dedicated efforts for exploration.

The closest one that is also inherently generative, compositional, and abstract is the Simplified version of the CommAI Navigation (SCAN) (Lake & Baroni, 2018), an instruction following task. Essentially, SCAN is seq2seq translation, with little uncertainty or variation in primitives. Hence, it does not test agents' perceptual generalization or algorithmic generalziation. In contrast, HALMA is a task for visual concept development and rapid problem solving. Agents need to understand concepts from visuomotor experience and make smart decisions to acquire utility.

## 6 GENERAL DISCUSSIONS

In spite of its synthetic nature, we believe HALMA is an impeccable testbed for rapid problem solving that resembles real-world ones. The dedicated design of its internal state facilitates in-depth and comprehensive analyses on agents' capacity in concept development, abstract reasoning, and meta learning that are otherwise impossible with existing problem-solving tasks. Agents can only pass the dynamically generated generalization tests if they possess adequate capacity to *understand* the abstract structure of this task and build a powerful solver upon this understanding. Our experiments demonstrate the inefficacy of model-free reinforcement learning agents in generalizing their understanding, even when incorporated with generic inductive biases. Towards this end, we would like to invite colleagues across the machine learning community to join our challenge.

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

CONTENTS

## A  PROBLEM SPACE OF HALMA

### A.1  VALIDITY OF OPTIMAL PATHS

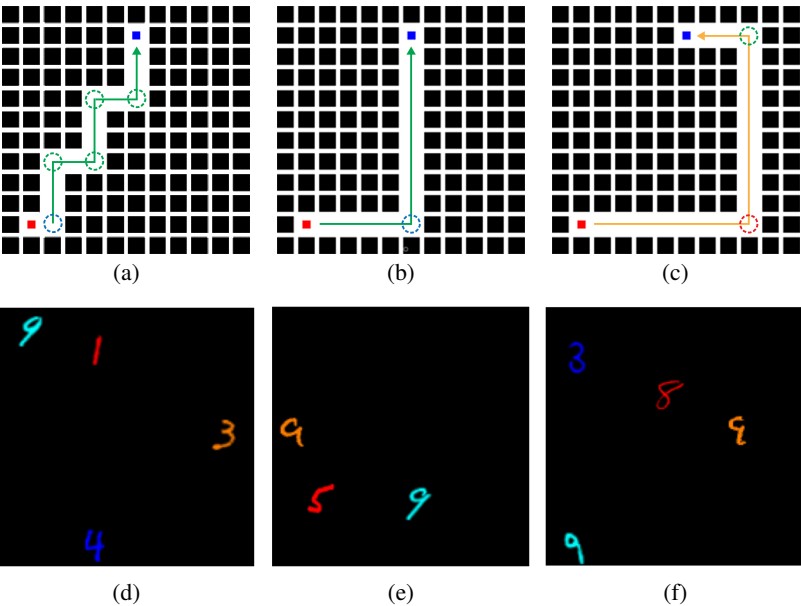

Figure S1: Examples of mazes and visual panels. (a) (b) Mazes have valid optimal paths in HALMA, highlighted in green and blue. (c) A maze configuration leads to ambiguous observations, highlighted by a red circle at the bottom-right corner. (d) (e) Visual panels correspond to the corner highlighted in blue in (a) (b), respectively. (f) Visual panels corresponds to the corner, highlighted in red in (c).

A common method to generate mazes in the grid-world is to (i) use randomized Prim's algorithm to create a connected area in the grid, and (ii) decide the positions of initial state and goal state, which naturally produce an optimal path between them. However, these randomly generated mazes may lead to ambiguous observations in HALMA and hinder the agent's formation of a coherent strategy. Specifically, let us look at the example mazes as shown Fig. S1; all three configurations could be generated by the Prim's algorithm. Mazes in Fig. S1 (a) (b) indeed produces valid optimal paths in HALMA. Unfortunately, the maze in Fig. S1 (c) leads to ambiguous observations, highlighted by red circle at the bottom-right. Below, we analyze them in-depth one by one. Recall that in HALMA, the agent observes a visual panel that contains several MNIST digits indicating the surrounding maze layout and the position of goal state. At the bottom-right in Fig. S1 (a), the agent may observe visual panels as in Fig. S1 (d), wherein a *1* and a *3* indicate that there are walls 1-grid away to the **left ◀** and 3-grid away right **above ▲**. The agent can also know from *4* and *9* that the goal state is 4-grid away on the **right ▶** and 9-grid away **above ▲**. Hence, the agent should make an **affordable** move **towards the direction of goal**; in this case, it is ▲, the direction that *3* and *9* align on. The **same strategy would work** in all corners highlighted in green or blue circles in Fig. S1 (a) (b). However, **the strategy may fail** at the bottom-right corner in Fig. S1 (c), wherein the agent may observe a visual panel depicted in Fig. S1 (f). This visual panel contains a *8* and a *9*, indicating that there are walls 8-grid away to the **left ◀** and 9-grid away **above ▲**. The visual panel also includes a *3* and a *9*, indicating that the goal state is 3-grid away to the **left ◀** and 9-grid away right **above ▲**. By adopting the **very same strategy** described above, the agent would be able to choose either ◀ or ▲. However, based on the global map, we know that only ▲ is correct.

To eliminate the aforementioned ambiguity, instead of first generating the complete maze and then producing the optimal path, our solution is to reverse this generation process, *i.e.*, first generating the *valid* optimal path that rules out the ambiguity and then adding deceptive branches to construct a grid-world maze. Formally, a path is considered *invalid* if an agent possessing an oracle understanding of the concept space and acting in accord to the above **strategy** fails to make proper decisions. We find that valid optimal paths can typically be divided into 'L'-shaped segments (see Fig. S1 (a) (b)), whereas invalid paths are commonly 'C'-shaped, rendering the observations at one of their corners ambiguous. In short, each single step on the **valid optimal path** from the initial state position to the goal state position should reduce the Manhattan distance to the goal state position by 1.

## A.2 GENERATING MAZE PROBLEMS

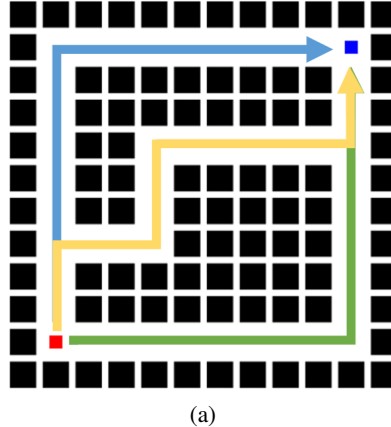
(a)

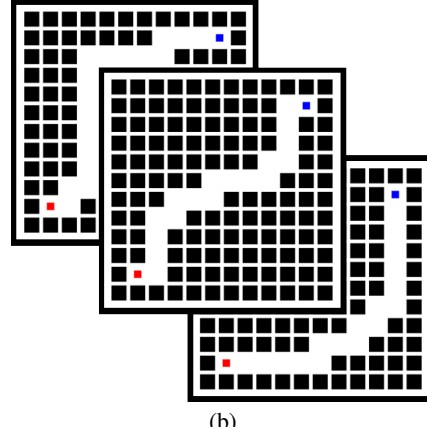
(b)

Figure S2: Illustration of valid optimal path generation. Given the initial state position and the goal state position, one can determine the directions where the optimal path should expand towards. For example, (a) Based on the initial state position, the optimal path can only expand upwards or to the right to reach the goal state position. (b) Examples of possible valid optimal paths.

After clarifying the validity of optimal paths, we are able to build a pipeline to automatically generate the desired mazes. Assuming that the position of the initial state is on the bottom-left to the position of the goal state (see an example in Fig. S2), the optimal path should only expand upwards or to the right to reach the goal state position. Hence, given the horizontal offset $m$ and vertical offset $n$ from the initial state position to the goal state position, there should be $C(m+n-2, n-1)$ valid optimal paths in total. Note that in HALMA, although all the positions of the initial state and the goal state are restricted within a $10 \times 10$ grid, it is able to produce $738,980$ possible optimal paths, exhibiting a rich and immense problem space in HALMA.

Next, we uniformly sample the optimal path from the maze set and add deceptive branches to these optimal paths. To maintain the validity of optimal path, we add a hint (*i.e.*, □, ○, △, or ⬠) at each T-junction and crossing to indicate the direction the agent should move towards. In theory, the deceptive branches can be arbitrarily complex as they do not influence the validity of the optimal path. To test whether an agent understands the concept of these hints and successfully transfers the learned knowledge to novel problems, we set the average *depth* of deceptive branches to $2$ in the training set and $5$ in the testing set. To provide sufficient training data for an agent to recognize these hints, we set the average *branching number* to $5$ in the training set.

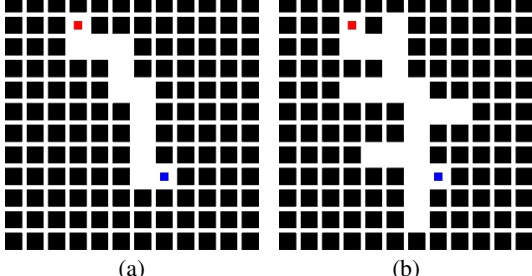
(a)      (b)

Figure S3: An example of adding deceptive branches to the valid optimal path.

## A.3 AN EXAMPLE TRIAL

In this section, we visualize an example trial completed by the *oracle* agent to further illustrate HALMA. The example trial is finished in $8$ steps; consecutive frames are shown in Fig. S4. Below, we provide detailed explanation of how the oracle agent makes its decision at each step:

(a) The oracle agent is spawned at an initial state position, highlighted by the red dot in the maze panel in Fig. S4 (a). Its observation is the visual panel, consisting of MNIST digits and a △ hint. Recall that the ground-truth semantics of △ indicates that the agent should move to the right, *i.e.*, ▶. Therefore, the agent who understands the meaning of △ would only need to know the distance to the wall and to the nearest T-junction or crossing[7] to the right in order to decide which action to take. Finally, recall that the yellow color is connected with ▶; the agent needs to make a comparison between the ⪲ and the 5, and chooses the lesser digit (*i.e.*, 2) as the distance it moves ▶.

---

[7]We will use the term *crossing* to refer to either of them henceforth, as well as in the main text.

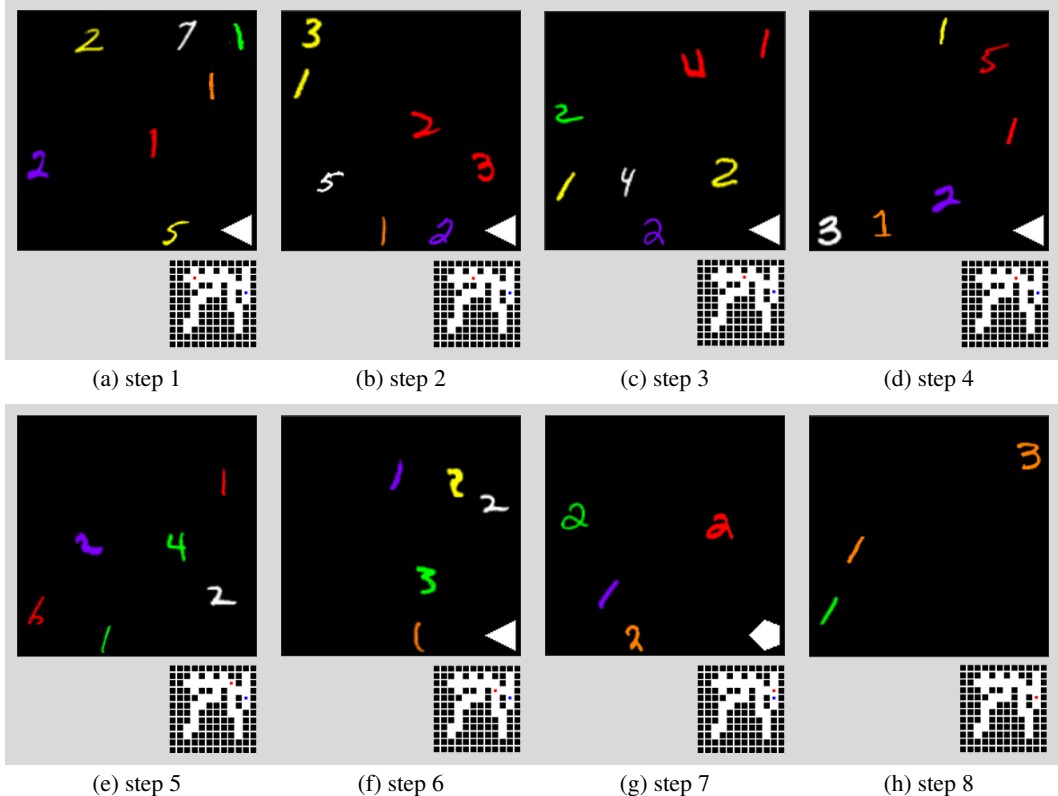

Figure S4: Visualization of an example trial completed by the oracle agent in 8 steps. Mazes at the bottom-right in (a)-(h) illustrate the trajectory of the oracle agent.

(b)-(d) In these frames, the oracle agent takes similar actions as in Fig. S4 (a). Note that in Fig. S4 (d), there is only one `yellow` color MNIST digit (*i.e.*, $\it{1}$ ) in the visual panel; therefore the agent may not need to make comparisons between digits. Hints $\triangle$ appear in all these visual panels since the oracle agent always stops at crossings.

(e) The oracle agent does not observe any hints for direction (*i.e.*, $\{\square, \bigcirc, \triangle, \diamondsuit\}$) in the visual panel because it is not at a crossing, therefore it needs to reason from the observation to decide which direction to move towards. The $\it{2}$ and the `white` digit 2 indicate that the goal state position is 2-grid below and 2-grid to the right. Additionally, the agent also observes no `yellow` digit in the panel, which indicates that the agent's current position is against the wall to the right. Therefore, the agent should move downwards. Finally, recall that the `green` color is connected with $\blacktriangledown$; the agent needs to make a comparison between the $\it{1}$ and the $\it{4}$, and chooses the lesser digit (*i.e.*, 1) as the distance it moves $\blacktriangledown$.

(f)-(g) The oracle agent takes similar actions in these frames as in previous frames. Note that in frame (g), the hint is $\diamondsuit$, which indicates that the agent should move downwards (*i.e.*, $\blacktriangledown$). Additionally, the $\it{1}$ and $\it{2}$ indicate that the goal state position is downwards 1-grid away, and there is no obstacle in the way until 2-grid away. The oracle agent can infer that it should move downwards $\blacktriangledown$ by 1 step to reach the goal state position.

(h) This frame shows the goal state in this trial. The example trial ends at this frame.

# B  FORMAL DEFINITIONS OF CONCEPT SPACES

## B.1  PRELIMINARY

For the sake of formalism, we borrow the terminology from the *General Pattern Theory* (Grenander, 1993). In case readers are not familiar with the General Pattern Theory, it is a mathematical study of *regular structures — configuration spaces, patterns* to account for the *combinatory* principle of our world. Adopting the language of abstract algebra, Grenander calls the basic unit of a regular structure/configuration space a *generator*, generically denoted as $g_i$. Any $g_i$ is associated with a number of *bonds* $\beta_j$, whose *value* $\beta_j(g_i)$ shall be within the *bond value space* $B$. Generators are

combined together by *connectors*. A connector $\sigma$ is a graph, say with $n$ sites. When $n$ generators are placed on a connectors' sites, we have a *configuration*, $c = \sigma(g_1, g_2, ..., g_n)$, which comes together with a set of *bond relations* $\rho : B \times B \rightarrow \{\text{TRUE}, \text{FALSE}\}$. A configuration is called *regular* if all bond relations return TRUE.

Despite of its generality, the formal language used by Grenander might appear somewhat abstract or peculiar to researchers in our community. Hence, we further elaborate below, from the perspective of *grammar*. A grammar is a regular structure, mostly studied in the community of natural language or linguistics to elucidate the combinatorial expressiveness in generating an immense set of configurations by composing only a considerably smaller set of words, using production rules. To account for the similar compositional and hierarchical nature in visual scenes, Zhu & Mumford (2007) introduced a stochastic grammar to the community of vision. They proposed an image grammar in an *And–Or Graph* (AOG) representation, where each Or-node points to alternative sub-configurations, and each And-node is decomposed into a number of sub-components. An AOG represents (i) the hierarchical decompositions from scenes to primitives and pixels, via non-terminal and terminal nodes, and (ii) the contexts for spatial and functional relations by horizontal links among the nodes. Below, to make this appendix self-contained, we summarize some key definitions:

**Definition 1** (Vocabulary). The vocabulary $V$ is a set of generators $g_i(\alpha_i)$, each associated with its bonds, $\beta_i = (\beta_{i,1}, ..., \beta_{i,d(i)})$. $\alpha_i$ is a vector of *attributes*. For instance, a visual generator may contain material properties of an object or the gender of a person as its attributes.[8] Bonds need to be connected with other bonds to form *attributed relations*; see the next definition.

**Definition 2** (Attributed Relations). Given an arbitrary set of generators $V$, a binary relation is a subset of the product set $V \times V$

$$\{(u, v)\} \subset V \times V.$$

An attributed binary relation is an augmented binary relation with a vector of attributes $\sigma$ and $\rho$

$$E = \{(u, v; \sigma, \rho) : u, v \in V\},$$

where $\sigma(u, v)$ represents the *connector* that binds $u$ and $v$, and $\rho(s, t)$ is a real number measuring the compatibility between $u$ and $v$. Then $\langle V, E \rangle$ is a graph, expressing the generalized relation $E$ on $S$. It is the *relation* that you are familiar with in object-oriented language such as First-Order Logics. For instance, the distance between two objects is an attributed relation. A $k$-way attributed relation is defined in a similar way as a subset of $V^k$.

**Definition 3** (Configuration). A configuration $C$ is a one-layer graph, often flattened from its hierarchical representation

$$C = \langle V, E \rangle.$$

For a visual scene, it is a spatial layout of entities in a scene at certain level of abstraction.

**Definition 4** (Parse Graph). A *parse graph* $pg$ consists of a hierarchical *parse tree* (defining "vertical" edges) and a number of relations E (defining "horizontal edges"):

$$pg = \langle pt, E \rangle.$$

The parse tree $pt$ is also an *And-tree*, whose non-terminal nodes are all And-nodes. The decomposition of each And-node $A$ into its parts is given by a *production rule*, which now produces not a string (like in natural language or linguistics) but a configuration:

$$\sigma : A \rightarrow C = \langle V, E \rangle.$$

A production should also associate the open bonds of $A$ with open bonds in $C$. The whole parse tree is a sequence of production rules:

$$pt = (\sigma_1, \sigma_2, ... \sigma_n).$$

The horizontal links $E$ consists of a number of directed or undirected relations among the *terminal* or *non-terminal* nodes:

$$E = E_{r_1} \cup E_{r_2} \cup ... \cup E_{r_k}.$$

These relations can be spatial relations, semantic relations, affordance relations, and causal relations. A parse graph $pg$, when *collapsed*, produces a series of flat configurations at each level of abstraction/detail:

$$pg \implies C.$$

---

[8]In computer vision, attributes are some properties of objects or agents that tend to remain the same.

**Definition 5** (And-Or Graph). An And–Or Graph is a 6-tuple for representing an grammar $\mathcal{G}$.

$$G = \langle S, V_N, V_T, \mathcal{R}, \Sigma, \mathcal{P} \rangle.$$

$S$ is the root node of a scene, $V_N = V^{\text{and}} \cup V^{\text{or}}$ is a set of non-terminal nodes, including an And-node set $V^{and}$ and an Or-node set $V^{or}$. The And-nodes plus sub-graphs formed by their children are the productions, whereas the Or-nodes are the vocabulary items. $V_T$ is a set of terminal nodes, for instance, visual primitives, parts, and objects. $\mathcal{R}$ is a number of relations between the nodes, $\Sigma$ is the set of all valid/regular configurations derivable from the grammar, *i.e.*, its language. $\mathcal{P}$ is the probability model defined on the And–Or Graph.

In sum, as a generic representation, an And-Or Graph can represent the hierarchical and relational knowledge of a visual scenario.[9] In the following subsections, we concretely define the *configuration space* of HALMA by grounding abstract notions in this subsection to specific components.

### B.2 CONCEPT SPACES OF HALMA

**Definition 6** (Axioms for Equivalence Relation =). An equivalence relation is a binary relation that is reflexive, symmetric, and transitive. For any generators $g_1$, $g_2$, and $g_3$:

- $g_1 = g_1$, (*Reflexivity*)
- $g_1 = g_2$ if and only if $g_2 = g_1$, (*Symmetry*)
- if $g_1 = g_2$ and $g_2 = g_3$, then $g_1 = g_3$. (*Transitivity*)

**Definition 7** (Axioms for Partial Order Relation $\leqslant$). A partial order is a binary relation that is reflexive, antisymmetric, and transitive. For any generators $g_1$, $g_2$, and $g_3$:

- $g_1 \leqslant g_1$, (*Reflexivity*)
- if $g_1 \leqslant g_2$ and $g_2 \leqslant g_1$, then $g_1 = g_2$, (*Antisymmetry*)
- if $g_1 \leqslant g_2$ and $g_2 \leqslant g_3$, then $g_1 \leqslant g_3$. (*Transitivity*)

**Definition 8** (Addition on Nature Numbers +). Given $\mathbb{N}$ and its *successor* function $s$ by *Peano Axioms*, we may have a *group* $\langle \mathbb{N}, + \rangle$ if we define *addition* + as: for $n, m \in \mathbb{N}$,

- $n + 0 = n$,
- $n + s(m) = s(n + m)$.

**Definition 9** (Subtraction on Nature Numbers −). Given $\mathbb{N}$ and $\leqslant$,

- let $m, n \in \mathbb{N}$, such that $m \leqslant n$;
- let $p \in \mathbb{N}$, such that $n = m + p$.

We define *subtraction* − as $n - m = p$.

**Definition 10** (Spatial Grammar of HALMA). The spatial grammar of HALMA is an Spatial And–Or Graph (S-AOG), which is a 6-tuple

$$G_S = \langle S_S, V_{N_S}, V_{T_S}, \mathcal{R}_S, \Sigma_S, \mathcal{P}_S \rangle,$$

where $S_S$ is the root node that represents the set of all visual panels, thus an Or-Node connected to nodes in $V_{N_S}$. There is only one element $v$ in $V_{N_S}$, representing an instance of visual panel. $v$ is a Set-Node since the number of digits in the panel may vary with different state; recall that it is because *zero* does not appear in the panel. $v$ produces all MNIST digits $\mathtt{d_i}$ (or hints) in the panel; it is a *composed concept*. These MNIST digits consist the terminal node $V_{T_S}$. They are *attributed* with $\mathtt{color}$, $\mathtt{scale}$, $\mathtt{location}$, $\mathtt{indication}$, and $\mathtt{category}$. Specifically, $\mathtt{color} = \{\mathtt{red, orange, yellow, green, cyan, blue, purple, white}\}$, and $\mathtt{indication} = \{\mathtt{wall} \vee \mathtt{crossing, goal}\}$. Ideally, the visual panel contains all nature numbers, $\mathtt{category}^* = \mathbb{N} \cup \{\bigcirc, \triangle, \square, \diamondsuit\}$. Currently, however, we only consider $\mathtt{category} = \{1, 2, 3, 4, 5, 6, 7, 8, 9\} \cup \{\bigcirc, \triangle, \square, \diamondsuit\}$. There is a *bijection* between $\mathtt{color}$ and $\{\blacktriangle, \blacktriangledown, \blacktriangleleft, \blacktriangleright\} \times \mathtt{indication}$, which gives rise to a *partition*, $\mathtt{type}$, over $V_{T_S}$. As terminal nodes, $V_{T_S}$ are *atomic generators*, hence *primitive concepts*. Though there can be many possible relations between these generators (*e.g.*, distance between MNIST digits, ordering of $\mathtt{scale}$ between MNIST digits), only the (*strict*) *partial order* over $\mathtt{category} \times \{\blacktriangle, \blacktriangledown, \blacktriangleleft, \blacktriangleright\}$, *i.e.*, $\langle \mathtt{S}, < \rangle$ is crucial to the task of HALMA. The definition of $\langle \mathtt{S}, < \rangle$ would come clear once we define $\mathtt{S}$ and how the concept of $\mathbb{N}$ is *bootstrapped* and grounded to $V_{T_S}$. $\mathcal{P}$ depends on the underlying maze problem since the valid configuration space $\Sigma_S$ of this grammar is all descriptions of states.

---

[9]Note that by *representation*, we do not necessarily mean how an artificial agent should represent such knowledge. Rather, it is a formalism for us humans to understand the internal structure of HALMA.

**Definition 11** (Semantics in HALMA). The *semantics* S in HALMA is a relation, a subset of $V_{T_S} \times$ category $\times \{\blacktriangle, \blacktriangledown, \blacktriangleleft, \blacktriangleright\}$

$$\text{S} \subset V_{T_S} \times \text{category} \times \{\blacktriangle, \blacktriangledown, \blacktriangleleft, \blacktriangleright\},$$

which is the ground-truth labeling of MNIST digits and their colors. For simplicity, we would slighly abuse this notion: In the remainder of the paper, we may regard S as a function $V_{T_S} \to$ category $\times \{\blacktriangle, \blacktriangledown, \blacktriangleleft, \blacktriangleright\}$ and also regard it as the *range* of this function.

**Definition 12** (Temporal Grammar of HALMA). The temporal grammar of HALMA is a Temporal And–Or Graph (T-AOG), which is a 6-tuple

$$G_T = \langle S_T, V_{N_T}, V_{T_T}, \mathcal{R}_T, \Sigma_T, \mathcal{P}_T \rangle,$$

where $S_T$ is the root node that represents the set of all options, thus an Or-Node connected to elements in $V_{N_T}$. Different from the spatial grammar, the temporal grammar has richer hierarchical structure, therefore there are more than one element in $V_{N_T}$, each representing an option opt. An option is a *composed concept*, which produces its constituting options/actions. The production rule $\rho$ is defined by the operation $+$. Production terminates when reaching terminal nodes $V_{T_T} = \{\square, \boxdot, \boxdot, \boxdot\}$. Since each of them are mapped to a semantic meaning (*i.e.*, moving 0, 1, 2, or 3 steps), they are *primitive concepts* of this grammar. All actions and options are attributed with $\{\blacktriangle, \blacktriangledown, \blacktriangleleft, \blacktriangleright\}$, which regularizes the production to be within the same type. Ideally, if we could build maze with infinite size, for each type, the production rule would specify a *group* over all nature numbers $\langle \mathbb{N}, + \rangle$. With that said, the only element in $\mathcal{R}_T$ is *equality* $=$. If we represent all elements in $\langle \mathbb{N}, + \rangle$ with sequences of primitive set along with equality over them, we have the valid configuration space $\Sigma_T$. $\mathcal{P}$ is the prior distribution of this numerical decomposition.

**Definition 13** (Affordance in HALMA). The *affordance* A in HALMA is a relation, a subset of $V_{T_S} \times (V_{N_T} \cup V_{T_T})$

$$\text{A} \subset V_{T_S} \times (V_{N_T} \cup V_{T_T}),$$

which is a partial $\geqslant$ relation between the semantics of atomic generators in the spatial grammar and all generators in the temporal grammar. It is a partial relation because defined within each type and its *inverse*. Namely, it is defined based on $\langle \mathbb{N}, +/-, \leqslant \rangle$. An action/option is *affordable* in a state if this relation returns true. Hence, affordance is a *bootstrapped concept* emerged from agents' interaction with the environment. Recall that there may be two MNIST digits with the same color in one panel; the lesser one indicates the distance till the nearest crossing, and the greater one indicates the distance to the wall. Regardless of their difference in semantics, both of them fit this definition well, though only the greater digit indicates the ground-truth affordance in the current maze.

**Definition 14** (Causal Structure of HALMA). The causal structure of HALMA is a Causal And–Or Graph (C-AOG), which is a 6-tuple

$$G_C = \langle S_C, V_{N_C}, V_{T_C}, \mathcal{R}_C, \Sigma_C, \mathcal{P}_C \rangle,$$

where $S_C$ is the root node that represents the set of all scenarios, thus an Or-Node connected to elements in $V_{N_C}$. $G_C$ links $G_S$ and $G_T$ together. Since the environment of HALMA is Markovian, we have $\Sigma_C \subset \Sigma_S \times \Sigma_T \times \Sigma_S$.[10] With that said, *generators* in the causal structure include $V_{N_S} \cup V_{T_S}$ and $V_{N_T} \cup V_{T_T}$. Namely, $V_{N_C} = V_{N_S} \cup V_{N_T}$; $V_{T_C} = V_{T_S} \cup V_{T_T}$. Definitions of production rules in the causal structure inherit from the spatial grammar and the temporal grammar. What uniquely defined here is $\mathcal{R}_C = \{\langle \text{S}, < \rangle \langle \text{A}, +/- . \leqslant \rangle, \langle \text{C}, +/-, = \rangle\}$. All these three relations are derivable from $\langle \mathbb{N}, +/-, =, < \rangle$. In the current setup of HALMA, $\mathcal{P}_C$ is deterministic. Reader who are familiar with symbolic planning may find the similarity between $G_C$ and STRIPS-style action languages (Fikes & Nilsson, 1971). Specifically, affordance A corresponds to the *precondition* of an action, whereas *causality* corresponds to the *effect* of an action, to be defined below.

**Definition 15** (Causality in HALMA). The *causality* C in HALMA is a relation, a subset of $V_{T_S} \times (V_{N_T} \cup V_{T_T}) \times V_{T_S}$

$$\text{C} \subset V_{T_S} \times (V_{N_T} \cup V_{T_T}) \times V_{T_S},$$

which is a partial $=$ relation between (i) the Cartesian product of the semantics of atomic generators in the spatial grammar and (ii) all generators in the temporal grammar and (iii) the semantics of atomic generators in the spatial grammar. Similar to the semantics in HALMA, we would somewhat abuse its notion and refer to it as a function $V_{T_S} \times (V_{N_T} \cup V_{T_T}) \to V_{T_S}$. Similar to A, it is a partial relation because defined within each type and its *inverse*. For domains where it is defined, its definition is based on $\langle \mathbb{N}, +/-, = \rangle$. It is also a *bootstrapped concept* emerged from interaction.

---

[10]Otherwise, causal configurations would be non-Markovian, $\Sigma_C \subset (\Sigma_S \times \Sigma_T)^* \times \Sigma_S$.

# C  STATISTICS OF VISUAL PANELS

Figure S5: Key statistics of visual panels in the HALMA training set. Each training set contains 100 HALMA grid-world mazes. We randomly sample 10 training sets and report the mean and standard deviation of the occurrence count of (a) colors, (b) number of digits in a panel, (c) digits distribution over these 10 sets, and (d) digits distribution in log-scale.

Recall that in HALMA, we use eight colors, *i.e.*, red, orange, yellow, green, cyan, blue, purple, and white, to specify the type of digits. Digits indicating the distance till a wall or the nearest crossing towards each direction (*i.e.*, ◀, ▲, ▶, and ▼) are colored red, orange, yellow, and green, respectively. Digits indicating the offset to the goal state are colored cyan, blue, purple, and white. Following the design of CLEVR (Johnson et al., 2017), in HALMA, we deliberately control the distribution of visual attributes, especially of COLOR, by sightly adjusting generated mazes to form a uniform distribution of digit type. Such design help to avoid possible strong biases in the data that agents can exploit to correctly take actions without reasoning. Below, we report key statistics of visual panels in the *training set* to demonstrate the uniformity of attributes distribution.

Fig. S5 (a) illustrates the color distribution of the visual panels. We produce an approximately uniform distribution for the color connected with distance to walls and crossings (*i.e.*, red, orange, yellow, and green) and for the color connected with offset to goal state (*i.e.*, cyan, blue, purple, and white) separately. We uniformly sample optimal paths and add deceptive branches when creating the mazes in the training set (see details in Appendix A) to form this distribution as an attempt to mitigate the color-conditional bias in the training set.

Fig. S5 (b) shows the distribution of number of digits in the panel. Number of digits in the panel are in an unimodal distribution. More than 90% panels in the training set has a number of digits between 3 and 6. Only <10% panels have 1-2 or 7-8 digits. No panel has a number of digits greater than 8.

Fig. S5 (c) (d) plot the distribution of digit in visual panels, revealing a long-tail distribution, where digit '1' has an occurrence number over 4,000, and digit '9' has an occurrence number less than 100. We consider this design as a nature of HALMA training set. Note that a greater digit tends to co-occur with the lesser digits in HALMA. For instance, if the agent passes a 9-grid-long passage step-by-step in a maze, it would observe not only the digit '9,' but also all the digits from '1' to '8.' Additionally, since we uniformly add branches on the optimal paths to create crossings, it adds to

the occurrence of lesser digits. In essence, this almost log-linear distribution aligns well with the natural distribution of digits or words for numbers in human language (Dehaene & Mehler, 1992).

# D  DYNAMICALLY GENERATE GENERALIZATION TESTS

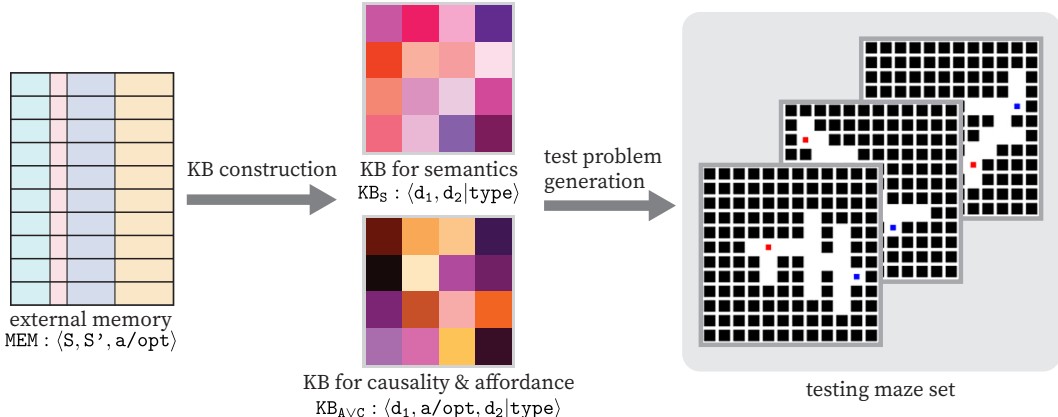

Figure S6: Illustration of the testing maze generation pipeline.

One of the unique features that HALMA possesses is its capability of pinpointing the model weaknesses by dynamically generating informative and definitive generalization tests according to agents' experience. During training, we save the running experience of the agent as its external memory MEM, specifically as a tuple, containing (i) a pair of states s and s', and (ii) the action/option a/opt the agent takes in this transition. Based on this external memory MEM, we build a pipeline that automatically generates the diagnostic testing set that tests a range of generalization abilities.

**Knowledge Base Construction**    As shown in Fig. S6, we first construct the Knowledge Base (KB) from the external memory MEM by converting the tuples to inequality hitmaps following these rules:

- If a pair of red, orange, yellow, or green digits $\langle d_1, d_2 \rangle$ occur in the same panel, then they are considered to represent the relation $\langle d_1, d_2 | \mathtt{type} \rangle$ that belongs to semantics inequality, where $d_1$ is the greater digit and $d_2$ the lesser one. Recall that the color mentioned above are connected with directions ▲, ▼, ◀, and ▶, respectively. In short, this KB is for $\langle S, < \rangle$.
- If the digit colored red, orange, yellow, or green changes in states s and s', and if both digits are non-zero, then they are considered to represent the relation $\langle d_1, d_2 | \mathtt{type} \rangle$ that belongs to affordance inequality and causality equality, where $d_1$ is the greater digit in s and s' and $d_2$ the lesser one. In short, this KB is for $\langle A, +/-, < \rangle \cup \langle C, +/-, = \rangle$.
- If the digit colored red, orange, yellow, or green appears in state s and disappear in s', meaning that the agent consumes the distance the digit d represents, we consider the indication of that digit is revealed to the agent. We can therefore consider that the relation $\langle d_1, d_2 | \mathtt{type} \rangle$ that belongs to affordance inequality is explored, where $d_1$ and $d_2$ are digits understood through affordance, and $d_1$ is the greater digit and $d_2$ the lesser one. In short, this KB is for $\langle A, +/-, = \rangle$.

**Test Problem Generation**    We pull inequality pairs from the constructed KB according to Section 3.4 to generate the testing mazes. Specifically, each inequality relation pair $\langle d_1, d_2 | \mathtt{type} \rangle$ contains a pair of digits $\langle d_1, d_2 \rangle$; we use the greater one as the distance till a wall, and the lesser one as the distance till a crossing. Therefore, we are able to incorporate the concepts to the maze layout and use the generated maze set to test the agent abilities of generalization.

As mentioned in Section 3.4, generalization test problems in HALMA are categorized into 3 different groups, *i.e.*, Semantic Test (**ST**), Affordance Test (**AfT**), and Analogy Test (**AnT**); there are also some more specific tests within each of these groups. Below, we provide detailed, concrete, and illustrative examples for each test unit:

- **ST**-1: We would like to know whether the agent can understand novel MNIST-digit-level combinations and make right decisions from those observations. Therefore, we pull inequality relation pairs that rarely co-occur in the external memory MEM and create ST-1 mazes based on these digit pairs. For instance, if the agent observed a combination $\langle 3, 5, 1, 7 \rangle$ during training, we would

like to test whether the agent can make right decisions given $\langle 3, 5, 4, 2 \rangle$, which is rarely or never observed during training, *i.e.*, $\langle 3, 5, 4, 2 \rangle \notin$ MEM. To achieve this goal, we can create a maze segment, where there is a crossing 3-grid away upwards and a wall 5-grid away in the same direction to ensure that the visual panel includes $\langle 3, 5 \rangle$. We then add a wall 4-grid away downwards to include the 4, and set the goal state position 2-grid away downwards to include the 2. Finally, we can assemble this kind of maze segments to create the desired testing mazes.

- **ST**-2: We would like to know whether the agent can recognize novel digit attributes combinations. We focus on the novel combination of `color` and MNIST `category` in **ST**-2 mazes. For instance, if the agent observes $\langle 3, 5 \rangle$ during training, we would test whether the agent can take right actions given $\langle 3, 5 \rangle$, which should be never seen during training, *i.e.*, $\langle 3, 5 \rangle \notin$ MEM. We can create a maze segment, where there is a crossing 3-grid away downwards and a wall 5-grid away in the same direction to ensure that the visual panel includes $\langle 3, 5 \rangle$.

- **Aft**-1: We would like to know whether the agent can understand the indication of MNIST digits through causal transitions in **Aft**-1. For instance, if the agent observed the 5 in the visual panel, moved 2 steps to the left ◀ : ⬚, and observed the 3, we would expect the agent to understand that $3 < 5$ through this transition. We can directly pull this kind of inequality pairs from $KB_{A \vee C}$. Note that to create pure testing mazes for **Aft**-1, we need to ensure that there are neither direct observations of the digit pair $\langle 3, 5 \rangle$ from visual panels, nor visual observations of $\langle 3, 5 \rangle$, $\langle 3, 5 \rangle$, or $\langle 3, 5 \rangle$, which would help to infer the inequality relation $3 < 5$ if the agent could recognize novel digit attributes combinations as in **ST**. In short, if an inequality pair $\langle d_1, d_2 \rangle$ is pulled from $KB_{A \vee C}$ to create the testing mazes for **Aft**-1, then we must have $\langle d_1, d_2 \rangle \notin KB_S$. We can then similarly create a maze segment as in **ST**.

- **Aft**-2: We would like to know whether the agent can understand the indication of MNIST digits through affordance in **Aft**-2. For instance, if the agent exploited the affordance of 5 with ▲ : ⬚ + ⬚ and exploited the affordance of 3 with ▲ : ⬚ during training, we would expect it to understand these two digits. We can directly pull such inequality pairs from $KB_A$ to create testing mazes for **Aft**-2. Note that the inequality pair $\langle d_1, d_2 \rangle$ pulled from $KB_A$ should not be in $KB_S$ for the same reason as in **Aft**-1.

- **Aft**-3 and **Aft**-4: We would like to know whether the agent can understand the indication of MNIST digits through transitions and affordance based on their understanding of the composition of visual attributes. For instance, if the agent's causality and affordance knowledge base $KB_{A \vee C}$ included the inequality pair $\langle 3, 5 \rangle$ or $\langle 3, 5 \rangle$, we would test whether the agent can understand $\langle 3, 5 \rangle$. Note that we need to ensure that $\langle 3, 5 \rangle$ is not in the $KB_S$.

- **AnT**: Note that in **ST** and **Aft**, we only test the direct inequality relation between digits. Here, we test the agent's understanding of transitivity of inequality relations in **AnT**. We expect agents to acquire the understanding of transitivity with analogical reasoning. For instance, if the agent's $KB_{AfT \vee ST}$ included a analogical template $\{\langle 3, 4 \rangle, \langle 4, 5 \rangle, \langle 3, 5 \rangle\}$, we would expect agents to learn analogical reasoning from this base case. If there was another pair of tuples $\langle 6, 7 \rangle, \langle 7, 8 \rangle$ in $KB_{AfT \vee ST}$, and further given that $\langle 6, 8 \rangle$ was not in the $KB_{AfT \vee ST}$, we would test the agent's understanding of transitivity from the analogical template.

Recall that in HALMA, we use 10 MNIST categories to indicate the distance till a wall or the nearest crossing, from which we extract the inequality relations and form the knowledge base. The number of inequality pairs is thus limited. Because the test units listed above are mutually exclusive, it is likely that some of the test problems may not be generated if the agent's experience, along with already generated tests, cover the full space of inequality. This explains the "-" in Table 1.

# E  DETAILS OF MODELS

## E.1  HYPER-PARAMETERS OF TD3

Table S1: Hyper-parameters of TD3

| Hyper-parameters | Value |
|---|---|
| Optimizer | Adam (Kingma & Ba, 2014) |
| Learning rate for actor | 1e-4 |
| Batch size | 128 |
| $\epsilon$ of Adam | 1e-8 |
| Discounting factor | 0.95 |
| Initial $\epsilon$ for $\epsilon$-greedy | 0.1 |
| Ending $\epsilon$ for $\epsilon$-greedy | 0.95 |
| Decay steps for $\epsilon$-greedy | 100,000 |
| Policy update delay | 5 |
| Target update rate | 0.995 |
| Replay buffer size | 10,000 |

## E.2  ARCHITECTURE OF AGENTS

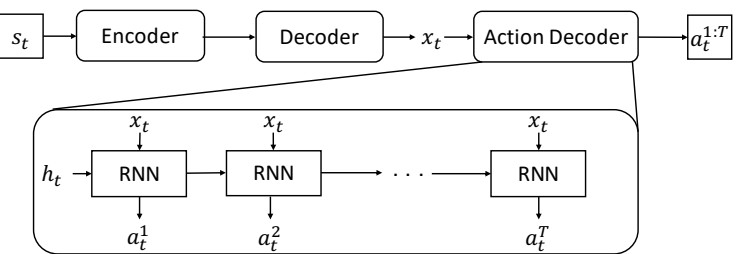

Figure S7: Architecture of the actor model, where T is equal to `max_opt_len`.

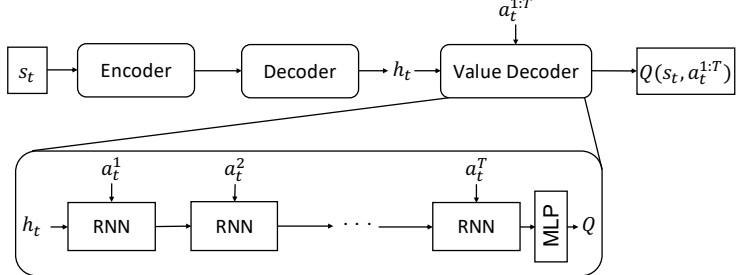

Figure S8: Architecture of the critic model, where T is equal to `max_opt_len`.

The overall architectures of the actor and the critic model employed by our agents are illustrated in Fig. S7 and Fig. S8, respectively. All agents share the same implementation of the action decoder and the value decoder, which allows them to work with action sequences, *i.e.*, option. Note that the hidden vector $h_t$ in the actor is simply initialized as a zero vector, and the critic uses the output of decoder instead to condition its output Q value on the state input.

The major difference among agents lies in the implementation of their inductive biases, *i.e.*, encoder and decoder. We provide a summary, along with some other hyperparemeters, in Table S2.

Table S2: Architectural parameters of evaluated agents

| Agent | Architecture |
|---|---|
| *Shared* | |
|     Nonlinearity | ReLU |
| *MLP Agent* | |
|     Encoder | MLP with hidden units [128, 128]. |
|     Decoder | None |
| *LSTM Agent* | |
|     Encoder | MLP with hidden units [128, 128]. |
|     Decoder | LSTM with layer normalization (Ba et al., 2016) and hidden units [128]. |
| *Transformer Agent* | |
|     Encoder | A stack of four multi-head self-attention layers, with hidden units [128], four heads and layer normalization, followed by a maximum pooing layer. Parameters are shared across all the attention layers (Zambaldi et al., 2019). |
|     Decoder | MLP with hidden units [128, 128]. |
| *Transformer+LSTM Agent* | |
|     Encoder | Identical to *Transformer Agent*. |
|     Decoder | MLP with hidden units [128, 128], followed by LSTM with layer normalization (Ba et al., 2016) and hidden units [128]. |
| *CNN Agent* | |
|     Encoder | CNN with kernel parameters [(3, 32, 6, 4), (32, 64, 6, 4), (64, 128, 7, 1)] (number of input filters, number of output filters, kernel size, and stride size by ordering). |
|     Decoder | MLP with hidden units [128]. |
| *CNN+Transformer Agent* | |
|     Encoder | CNN with kernel parameters [(3, 32, 4, 4, 0), (32, 64, 4, 4, 0), (64, 128, 3, 2, 1)] (number of input filters, number of output filters, kernel size, stride size, and padding size by ordering); resized to $4 \times 4$ slots, concatenated with positional embedding (Appendix F.1); followed by the encoder of *Transformer Agent*. |
|     Decoder | Identical to *Transformer Agent*. |
| *SPACE Agent* | |
|     Encoder | We adopt the original setup of SPACE (Lin et al., 2020) for the `image_encoder` and the `what_encoder`. We concatenate latent vectors for the shape ($Z_{what}$) and the presence ($Z_{where}$) of each object. In sum, there are $8 \times 8$ object slots, each is a 33-D vector. They are then fed to the encoder of *Transformer Agent*. |
|     Decoder | Identical to *Transformer Agent*. |

# F EXPERIMENTAL DETAILS

## F.1 TASK PARAMETERS AND EXPERIMENTAL PROTOCOL

**Task parameters** Task parameters for HALMA are mostly defined in Section 3.3, explicitly specified for the formulation of rapid problem solving. Table S3 summarizes these parameters.

Table S3: Task parameters of HALMA

| Task parameters | Value |
|---|---|
| Maximum #steps in an episode ($L$) | 500 |
| Maximum #trials in an episode ($N$) | 10 |
| Maximum #steps in a trial($H$) | 200 |
| Discounting factor ($\gamma$) | 0.95 |
| Goal reward ($R_g$) | 100 |
| Penalty on invalid action ($R_a$) | -5 |
| Penalty on invalid action ($R_a$) | -5 |
| Auxiliary rewards ($R_x$) | $1.0 \times (L_m(\text{agent}_{t-1}, \text{goal})$ $-L_m(\text{agent}_t, \text{goal}))$, where $L_m$ is the Manhattan distance. |

Given a set of generated HALMA problems, there is still one task parameter: `max_opt_len`, which is the maximum length of an option in one step. We tried three different setups, $\{1, 3, 5\}$. Intuitively, when `max_opt_len=1`, agents do not need to merge sub-options to improve planning efficiency, though they may still need to decide between $\{\square, \odot, \square, \odot\}$. With that said, the exploration and planning efficiency $\rho_p$ may be close to the optimal 1 as long as the ratio of goal reaching $\rho_g$ is high. In contrast, when `max_opt_len = 3` or 5, agents would need to understand the compositionality of the option space (*i.e.*, the temporal grammar) to improve $\rho_p$. In this case, as shown in Appendix F.2, most agents find it quite challenging to *plan optimally*. They may even get trouble in understanding affordance, hence have a higher ratio of invalid moves $\rho_p$ than when `max_opt_len=1`.

**Two types of observations** We provide two types of observations to the agents. One is a low-dimensional *symbolic observation space*. It represents the ground-truth MNIST digits, colors, and shape of hint symbols at crossing. Recall that in HALMA, the observation may have at most 10 MNIST digits[11] plus 1 crossing hint, and the value of digit range from -9 to 9,[12] which results in 10 one-hot vectors with an overall size of $11 \times 19$. For agents with permutation invariant modules (*e.g.*, transformers), we enforce the positional sensitivity by augmenting each one-hot vector with an extra indexing vector of size 10, which is essentially another one-hot vector that indicates the index. In our experiments, we observe that this index encoding is crucial to all the transformer-based agents.

We also offer a *visual observation space*, where the only observation is the visual panel of HALMA, as introduced in Sections 3.1 and 3.2. We downsample them to a RGB image with size $(128, 128, 3)$ and re-scaled to $[0, 1]$. Agents for this type of observations require visual modules, such as CNN or SPACE (Lin et al., 2020) as detailed in Appendix E.2.

**Training protocol** We generated 100 mazes for training. An ablation study on the volume of training set can be found in Appendix G. Each agent is trained for 2000 episodes under the task formulation introduced in Section 3.3. All of them converged at the end of training, as illustrated in their learning curves in Appendix F.2. We tried 5 different seeds during training and report the best result. Note that different from classical reinforcement learning tasks, where there is no explicit split for training and testing hence training curves are reported for quantitative evaluation, we provide training curves merely for justifying the validity of our training.

**Testing protocol** We test all agents in (i) the training problems, (ii) test problems generated by random split in the problem space, and (iii) test problems dynamically generated according to Section 3.4. The former two are provided mainly for reference. Interestingly, most agents perform almost equally well on these two, consistent with prior works (Guez et al., 2019; Cobbe et al., 2019). For all tests or dynamically generated subtests, we test with $\tilde{1}50$ mazes and summarize over 3 different seeds to calculate mean and standard deviation. A test is skipped if the dynamic generation fails, as introduced in Appendix D.

---

[11] 4 for crossings, 4 for distance to the walls, and the remainder, 2, for distance to the goal.

[12] For the two goal digits only, while others are only allowed to be in $\{0, 1, 2, 3, 4, 5, 6, 7, 8, 9\}$; the hint only has 4 different values)

### F.2   Learning Curves

To validate the convergence during training, we provide the learning curves of agents trained under different settings (mainly on the different choice of `max_opt_len`) in Figs. S9 to S11. We report the number of finished trials and the ratio of invalid actions in each training episode. The moving average (with a window size of the number of mazes in the training set) of these two metrics can reflect $\rho_g$ and $\rho_a$ in training. These curves suggest that all agents with symbolic observations converge before 2000 episodes in terms of the goal reaching rate and invalid action ratio. For the visual observation, however, agents struggles on both metrics when the action space is large (`max_opt_len=3` or `max_opt_len=5`). Their performances remain almost the same after 2000 episodes. Hence, we report the test results with `max_opt_len=1` in the main paper; full results can be found below.

Training (N = 10, up to 2000 episodes, maximum option length = 5)

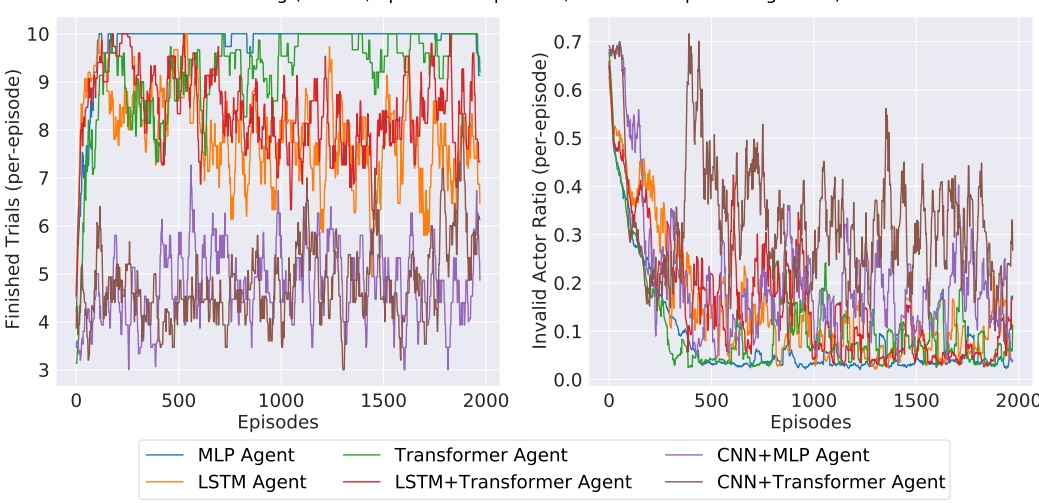

Figure S9: Learning curves of the evaluated agents with `max_opt_len=5`.

Training (N = 10, up to 2000 episodes, maximum option length = 3)

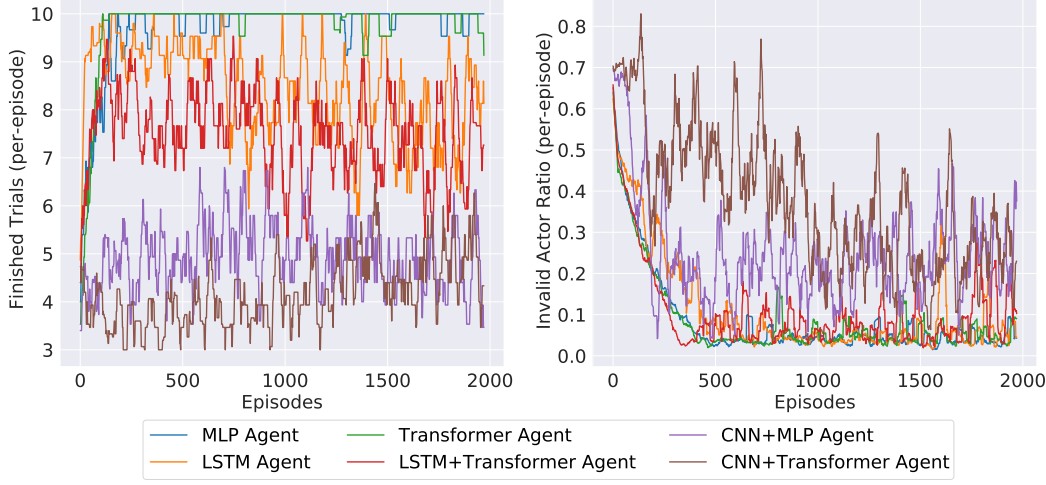

Figure S10: Learning curves of the evaluated agents with `max_opt_len=3`.

### F.3   SPACE Model

**Architecture and Hyperparemeters**   We adopt the original setup of SPACE (Lin et al., 2020) except for a simple modification in the background encoder. Specifically, we replace their `StrongCompDecoder` with their `CompDecoder`.

**Reconstruction, Segmentation and Detection on Testing Set**   We train the SPACE model with all visual panels in the training set. To qualitatively evaluate the generalization capability of the SPACE model, we visualize their inference results in a hold-out testing set; it is essentially a set of visual panels from randomly generated test problems; see Fig. S12 for an example. The SPACE model

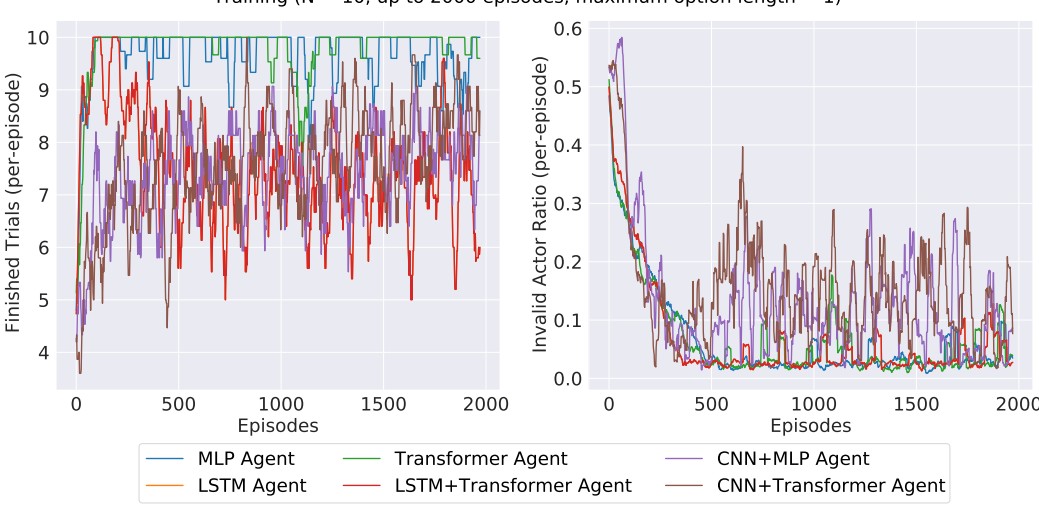

Figure S11: Learning curves of the evaluated agents with `max_opt_len=1`.

generalizes remarkably well in terms of reconstruction, detection, and segmentation, consistent with the original results reported by (Lin et al., 2020).

**Investigating the Latent Space**  We further investigate the efficacy of the SPACE model in disentangling independent latent factors from visual panels. Specifically, we adopt a standard methodology in the unsupervised disentanglement learning literature (Higgins et al., 2017), linear probing.

We train a linear SVM classifier using the latent representations of colored MNIST digits obtained from the encoder of the SPACE model. We observe that the output vector of SPACE encoder have multiple slots representing the objects (digits) in the input image, and that the connection between slots and input objects is implicit. Hence, we calculate the IoU of predicted bounding box and ground-truth bounding box to assign each slot to an input object as its semantic label. In this work, there are 64 slots in the output vector and no more than 11 objects in the input image. Therefore, it is likely that several slots are assigned to the same object. We save for each object only the slot with the maximum IoU to remove redundancy in the data and obtain $8,932$ 33-D latent vectors in total. We use 70% of these samples as training data and perform testing on the held-out 30% data by randomly splitting the latent vectors. We set the penalty parameter 'C' of SVM as 10 in all experiments and use balanced sampling when training the classifier. SVM classifier is implemented with the scikit-learn package (Pedregosa et al., 2011).

Table S4: Accuracy of `color` and MNIST `category` classification.

| Task | color | MNIST category |
|---|---|---|
| Acc. | 59.67$\pm$0.58 | 50.56 $\pm$0.92 |

We test the classification accuracy in terms of `color` and MNIST `category` and report the overall accuracy in Table S4. Each result is averaged over 10 random split of latent vectors. In addition, we provide the confusion matrix of these two attributes (Fig. S13) to illustrate the categorical accuracy. Results in Fig. S13 (a) demonstrates that the SPACE model performs relatively well on the first four colors, *i.e.*, `red`, `orange`, `yellow`, and `green`, while poorly on the rest. It partly explains SPACE agents' high invalid move ratio $\rho_a$ and low goal reaching ratio $\rho_g$ in HALMA, *i.e.*, agents cannot tell the correct direction. Results in Fig. S13 (b) demonstrates that the SPACE model does not handle the long-tail distribution of digits, and partly explains SPACE agents' high invalid move ratio $\rho_a$ and low efficiency ratio $\rho_p$ in HALMA, *i.e.*, agents do not know "what it is" in the first place.

## G  ADDITIONAL EXPERIMENTS

### G.1  ABLATION STUDY ON THE VOLUME OF TRAINING SET

The thesis argument of our work is that humanlike agents shall generalize their understanding under limited exposure to the underlying concept spaces. To further investigate how the degree of expo-

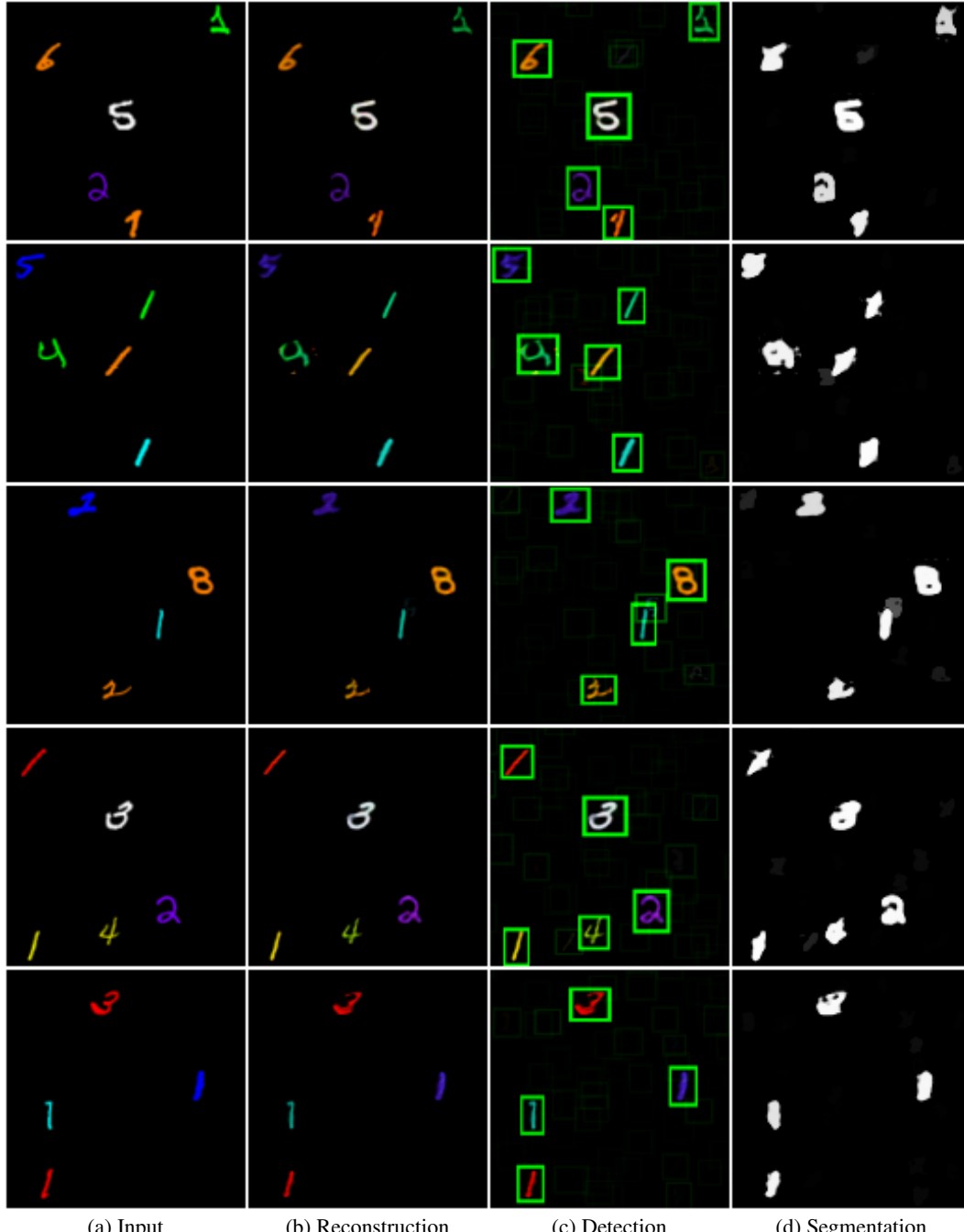

|                |                     |               |                  |
|:--------------:|:-------------------:|:-------------:|:----------------:|
| (a) Input      | (b) Reconstruction  | (c) Detection | (d) Segmentation |

Figure S12: Visualization of SPACE's reconstruction, detection, and segmentation on hold-out **testing set**.

sure would affect agents performance in HALMA, we first conduct an ablations study with different numbers of training mazes. Specifically, we experiment with four setups of the maze quantity for agents to explore during training: $100, 300, 500, 1000$ (results of 100 training mazes are reused from the main experiment as it is our default setting). Here we only evaluate agents with symbolic input: MLP agents, LSTM agents, Transformer agents and Transformer+LSTM agents. We report the three measures $\rho_a$, $\rho_g$ and $\rho_p$ with all the testing protocols (training problems, problems from random split in the problem space and dynamically-generated testing problems) in Fig. S14. Note that measures in dynamically-generated tests are merged across subtests for better comparison.

The results read that, all agents could gain a performance boost with increased exposure during training. Specifically, there is a significant promotion for the metric of goal reaching rate $\rho_g$ in the

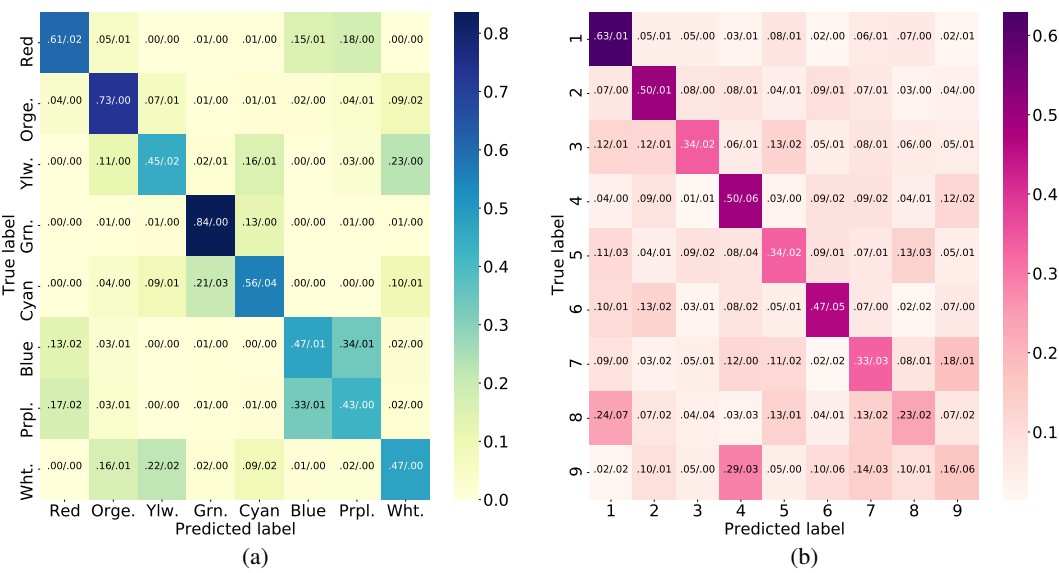

Figure S13: Confusion matrix of linear classifier trained on latent representations from SPACE encoder. Confusion matrix of (a) `color` and (b) MNIST `category` are measured and averaged over 10 random split testing set. Mean value and standard deviation of the accuracy are displayed in the matrix.

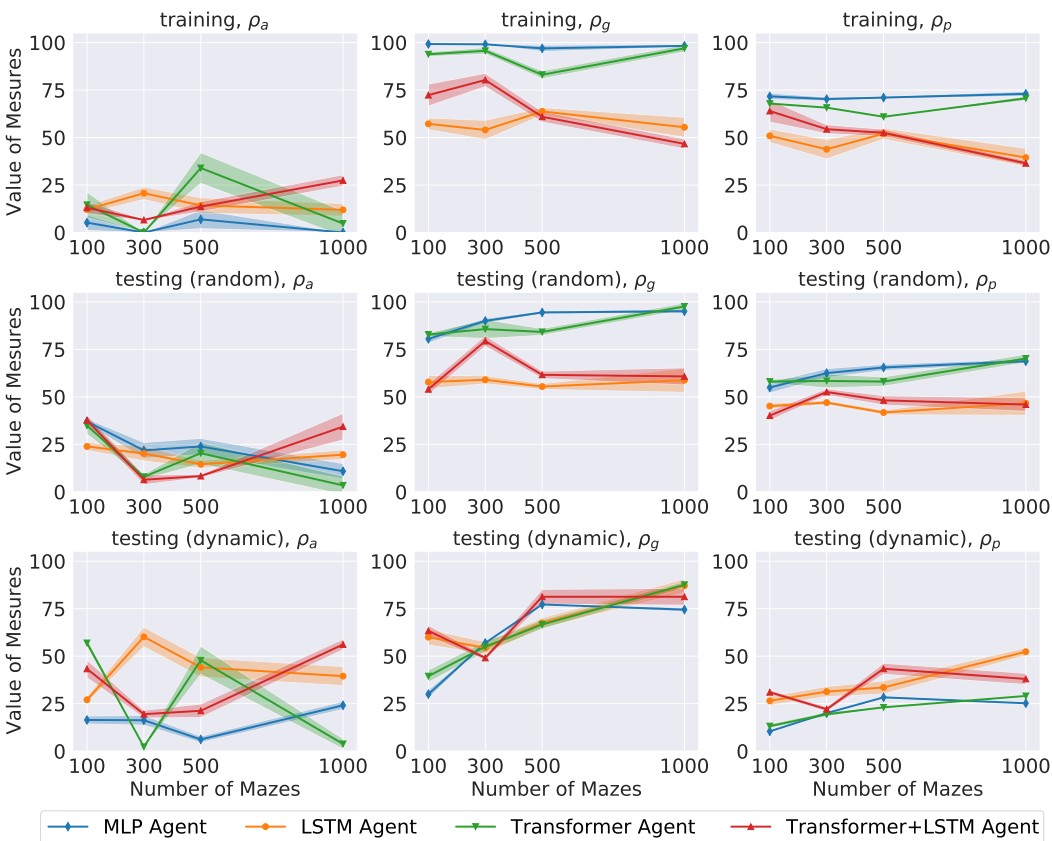

Figure S14: Ablation study of different number of training mazes.

challenging dynamic testing (from 30-60% to 80%). More interestingly, starting from 300 training mazes, the distinction between different inductive biases vanishes. While the efficiency ratio $\rho_p$ could also benefit from increased exposure, it reaches only around 50% at best. As for the ratio of invalid moves $\rho_a$, even though it reaches around 10% in random split for stateless agent when trained with 1000 mazes, no clear trend can be detected in dynamic testing overall, which may

suggest agents' limitation in understanding affordance with the temporal grammar or under the long-tail distribution of digits.

## G.2 ABLATION STUDY ON THE MAXIMUM OPTION LENGTH `MAX_OPT_LEN`

Our design to include the notion of option challenges agents' understanding in the temporal grammar and the causal structure. To further illustrate the difficulty of this specific challenge, we also perform an ablation study on three setups of maximum option length `max_opt_len`. In general, agents' performance degrades on all metrics with `max_opt_len` increases. In particular, the ratio of invalid moves $\rho_a$ increases and the efficiency ratio $\rho_p$ drops significantly since `max_opt_len=3` in dynamic testing, suggesting that agents all have hard time understanding either the temporal grammar or the causal structure of HALMA. These results validate our argument that significant efforts are still in need for humanlike abstraction learning. Therefore, we choose to make the length of 5 as our default setting in the main paper so as to make HALMA a more challenging territory.

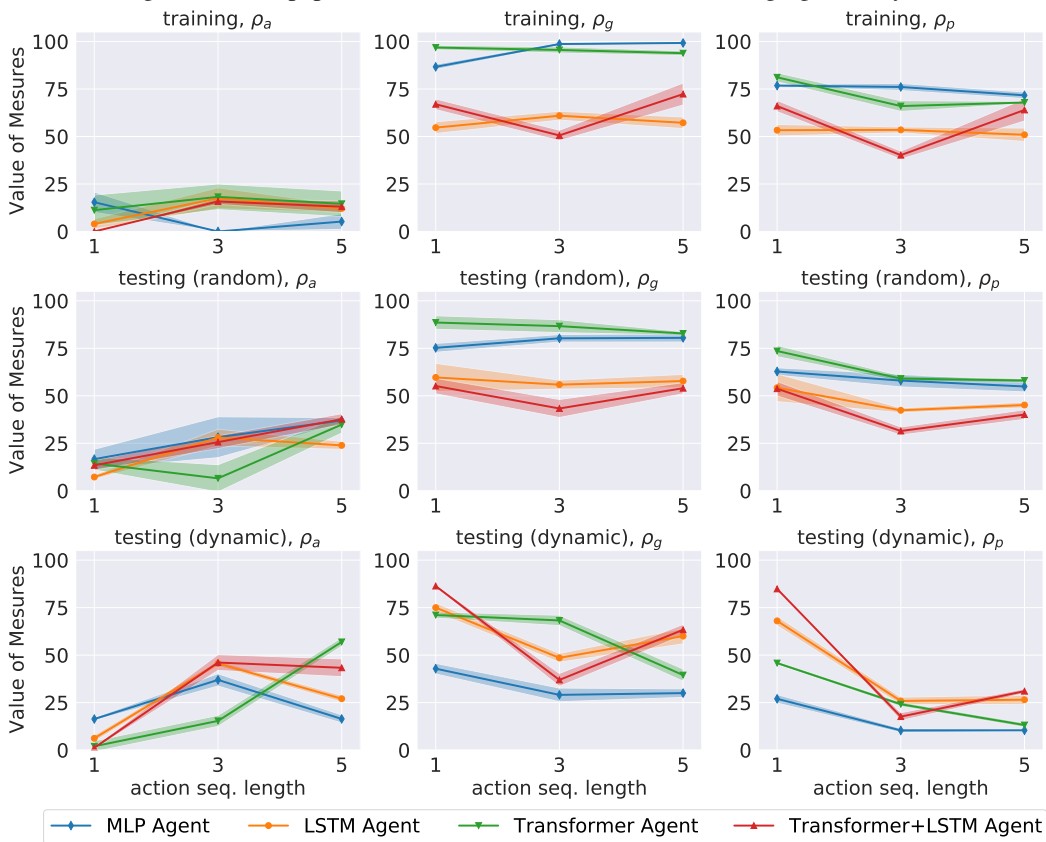

Figure S15: Ablation study of different `max_opt_len` (symbolic observations).

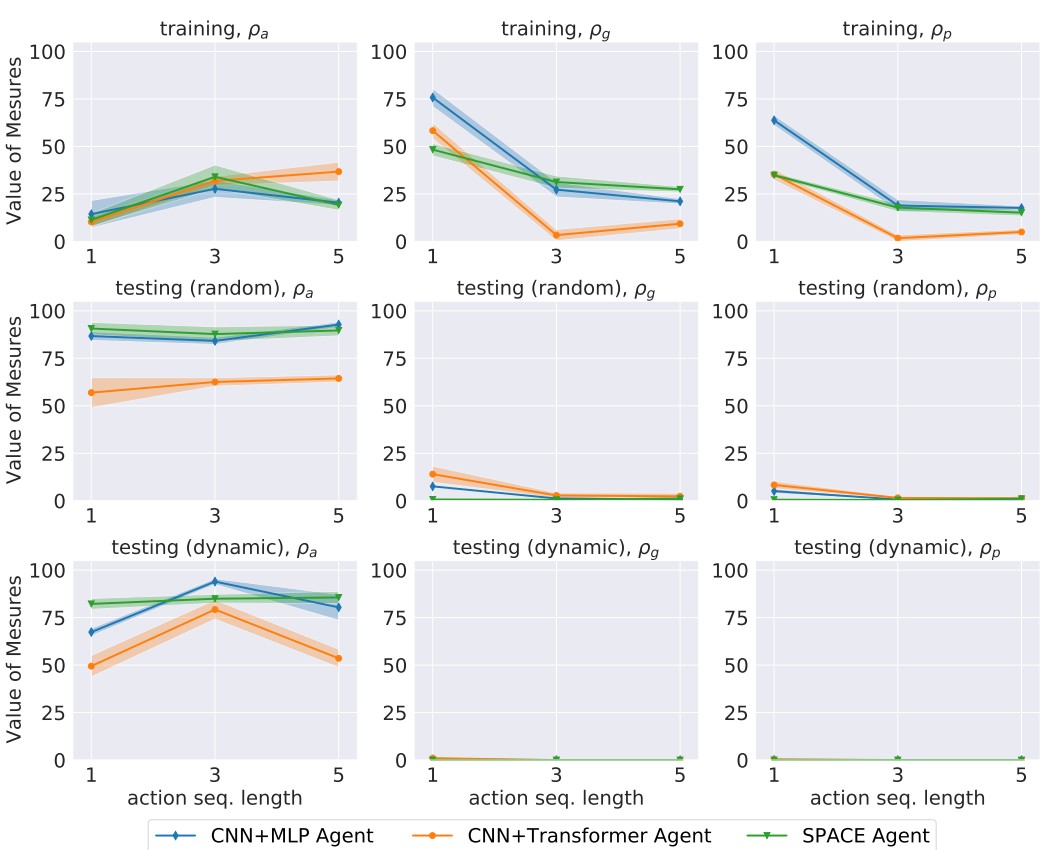

Figure S16: Ablation study of different max_opt_len (visual observations).

