# OpenReview forum: "HALMA: Humanlike Abstraction Learning Meets Affordance in Rapid Problem Solving"
_ICLR.cc/2021/Conference — Reject_

### Official Review · AnonReviewer4 · 2020-10-28
**Nice new benchmark**

**Rating:** 7
**Confidence:** 3

**Review:**

##########################################################################

Summary:

The paper introduces a new benchmark which measures agent reasoning abilities and their generalization of 3 kinds: perceptual, conceptual and algorithmic. The paper extensively motivates this benchmark and shows experiments with training RL agents on it.



##########################################################################

Reasons for score:

New benchmarks for measuring intelligence are important for driving the field. This one seems interesting.


##########################################################################

Pros:

1. New benchmark which tries to provide a comprehensive measure of generalization, which is a very important topic.
2. Paper is nicely written and illustrated - pleasure to read!
3. Benchmark/code will be published upon acceptance, so the whole community will be able to profit from it.

##########################################################################

Cons:

1. The history of artificial domains as AGI playgrounds has taught us a few bitter lessons. To give one very recent example, the winner of Abstraction and Reasoning Challenge (yeah, that one proposed by Francois Chollet to measure general intelligence https://arxiv.org/abs/1911.01547) wrote "Unfortunately, I don't feel like my solution itself brings us closer to AGI." https://www.kaggle.com/c/abstraction-and-reasoning-challenge/discussion/154597. The solution was some handcrafted search algorithm, no machine learning. To be fair, the challenge is very far from being solved - and yet the winning solution from 914 teams didn't teach us much. Is there anything in the design of HALMA which we expect will protect it from similar problems?

2. No human baselines are given. How do we know that our agents achieved human performance?

##########################################################################

Questions during rebuttal period:

I would be curious to know the authors' opinion on Cons above.

#########################################################################

Minor suggestions and typos:

(1) affordnace -> affordance

---

> ### Author Response · Authors · 2020-11-18
> **Response to R4 (1/2)**
>
> We thank R4 for acknowledging the proposed HALMA benchmark is "important for driving the field" and "interesting." We hope our detailed responses below would resolve the remaining questions you have and therefore increase the rating of our work.
>
> > To be fair, the challenge [ARC] is very far from being solved - and yet the winning solution from 914 teams didn't teach us much. Is there anything in the design of HALMA which we expect will protect it from similar problems?
>
> Thank you for bringing Francois Chollet's seminal work to this discussion. Our work does align well with most of the designing principles listed in II.3.2. However, the goal of "intelligence" in HALMA, even though we did not make a bold claim to "measure intelligence" in the manuscript, is not as "broad" or "general" as that in ARC. Specifically, in ARC, agents are assumed with the priors of all core knowledge, and it is the "fluid intelligence" that is required to achieve success. Our observation, however, is that (i) to combine core knowledge to "adaptive machines" itself is a challenging problem, (ii) and the notion of "fluid intelligence" is never grounded to any concrete knowledge to give us a sense of what it is. In that manuscript, Chollet seems hesitant in revealing concretely what is expected to be learned, because he thinks such a revelation may encourage developers to "hard-code a shortcut." A contradiction here is, he also wrote, "goal definitions and evaluation benchmarks are among the most potent drivers of scientific progress." Perhaps it is just the lack of ground-truth in the research goal that made the participants lost in some aimless search, making the ARC challenge not as rewarding as expected.
>
> We adopted a different and more realistic philosophy when we design HALMA. Instead of hiding the concept space from researchers, we formally define it and systematically describe how the sub-modules (STC) interact. Instead of endowing agents with all core knowledge priors, we only provide them with the minimum set: the number of 0, 1, 2, 3 and the direction of up, down, left, right. According to the core knowledge doctrine, these are the cognitive capabilities developed in the first 1-3 months of infancy [1]. In developmental psychology, how do we humans or even animals learn and do exact arithmetic over and beyond this minimum set is still an open question [1]. Instead of directly targeting "fluid intelligence," HALMA is more focused on how to learn structured abstraction, thus akin to the formation of the initial level of "crystallized intelligence."
>
> In short, what Chollet worried would not happen in HALMA because hardcoding an oracle agent is almost trivial: it can be done by simply maintaining a program for the natural number system. We have made it so apparent that we believe researchers who want to join our challenge would instead study something more interesting: What is the proper inductive bias to learn and represent this system? It is true that researchers may still be influenced by the revelation of the concept space. However, we believe that would be somewhat positive since the study of this concept space itself can lead to something more general. Note that from a mathematical viewpoint, the algebraic structure of natural numbers, as defined in Appendix B, is quite fundamental and universal; it is probably the first structure taught in the courses of discrete maths and mathematical analysis in college, which simply because it is the foundation of all fancier structures in maths, computer science, and AI. In abstract algebra, relations such as equivalence, ordering and operations such as addition are basic units to capture the regularity/patterns in our world [2]. Hence, the study of how machines develop HALMA's concept space may bring significant impact to the machine learning community.
>
> Since we picked a different philosophy in designing HALMA, our benchmark directly addresses some weaknesses listed in III.2 of Chollet's manuscript:
> * In ARC, "generalization is not quantified." In contrast, the clearly defined concept space in HALMA helps us define both several evaluation metrics and some test categories.
> * In ARC, "the evaluation format is overly close-ended and binary... Further, real-world problem-solving often takes the form of an interactive process where hypotheses are formulated by the test-taker then empirically tested, iteratively." In contrast, the task formulation of rapid problem solving in HALMA does provide a quantitative account of this fast learning process during testing.
>
> In sum, HALMA is the "possible alternative" described by Chollet as "Repurposing skill benchmarks to measure broad generalization" in III.3.1. The crisp concept space of HALMA makes it possible to evaluate agents in "substantially new" problems. And to better implement that, we propose to generate test problems dynamically, which we hope will inspire a significant shift in the evaluation paradigm, as commented by R3.

---

> > ### Author Response · Authors · 2020-11-18
> > **Response to R4 (2/2)**
> >
> > > No human baselines are given. How do we know that our agents achieved human performance?
> >
> > We thank your suggestions. However, the design of human baselines that **precisely** mimic HALMA is not trivial and deserves a dedicated paper reviewed by experts in cognitive science, which we reserve for our future work. Unfortunately, our subject pool is closed due to COVID-19, so we may not report the **naive** human study results during this review process. Nevertheless, we actively seek collaboration with developmental physiologists to develop a more scientific plan for a fair comparison between machines and humans. Below, we provide established evidences that endorse our claims.
> >
> > Even though the capability of doing 1-digit arithmetic may not be developed in the earliest 1-3 months, it is indeed developed relatively early in childhood, highly possible with the assistance of language [1]. This brings some intellectual challenges to the design of the human study. As written by Chollet in the manuscript, "to make sure that humans test-takers do not bring further priors to the test, the test tasks should not rely on any acquired human knowledge (i.e., any knowledge beyond innate prior knowledge). For instance, they should not rely on language or learned symbols (e.g., arrows)." We can indeed design a naive human study that reduces this prior by replacing MNIST digits with Omniglot characters. However, since most adults are literate and can do arithmetic or counting quite robustly, the result from such human study is unfair to agents who are not assumed with language in HALMA.
> >
> > [1] Dehaene, The Number Sense: How the mind creates mathematics, 2011
> >
> > [2] Grenander, General pattern theory: A mathematical study of regular structures, 1993

---

> > > ### Comment · AnonReviewer4 · 2020-11-19
> > > **Response to authors**
> > >
> > > Thanks for the explanation! This part
> > >
> > > > hardcoding an oracle agent is almost trivial: it can be done by simply maintaining a program for the natural number system
> > >
> > > sound a bit worrying to me and again warrants a human baseline so that the readers can clearly understand the complexity of the proposed problems. It could also facilitate wider adoption of the benchmark: when people see that the best method produces X% but the human baseline is Y% >> X%, they think "aha, that's a benchmark where large gains are possible, we could make a breakthrough here". It is fine to do this later though, given the circumstances.

---

### Official Review · AnonReviewer1 · 2020-10-29
**Naive review**

**Rating:** 5
**Confidence:** 2

**Review:**

Disclaimer: Apologies I have very little background in the area of this paper and should have probably opted out. However I did read the text twice with interest.

The authors propose a task HALMA which involves a grid-world maze that is partially observed, and includes visual panels that contain conceptual reasoning tasks which the agent must solve to find an optimal path. The aim of this environment is to test three proposed 'levels of generalisation': 'perceptual', 'conceptual' and 'algorithmic'. The authors propose several tests for generalisation: 'semantic', 'affordance', 'analogy'. The authors propose a dynamic manner of evaluating the agents by showing them problems that they appear to not yet have demonstrably understood. The authors conclude that existing agents are woefully inadequate at these generalisation tests and invite researchers to approach this novel task.

I found the paper had an enjoyable positive style of discourse and built up the problem area quite nicely in the introduction, but was unfortunately very verbose and many important details were relegated to the appendix. For example the actual algorithm used to dynamically produce test-set elements was relegated to Appendix D --- however this seemed to be one of the central contributions to the paper. Furthermore the environment of HALMA was not very succinctly described. Possibly this paper has too much information for a conference submission and should fit better in a journal, otherwise it should be edited down to make room for a succinct description of the environment, and the evaluation approaches *in the main text*.

I was curious to what effect many of the research findings depended on the arbitrary size of the training set; and would have been interested to see some generalisation metrics reported as a function of training set size (100 mazes seems very small). I would have also liked to see a human baseline and then to have had human-normalised scores.

---

> ### Author Response · Authors · 2020-11-18
> **Response to R1 (1/2)**
>
> We thank R1 for constructive suggestions, despite out of the expertise. We understand that the trade-off (detailed below) may induce a heavier burden to R1 given limited review time. We sincerely thank you for your patience! We hope our detailed responses below would resolve the remaining questions you have and therefore increase the rating of our work.
>
> > [T]he paper had an enjoyable positive style of discourse and built up the problem area quite nicely in the introduction, but was unfortunately very verbose and many important details were relegated to the appendix. For example[,] the actual algorithm used to dynamically produce test-set elements was relegated to Appendix D --- however[,] this seemed to be one of the central contributions to the paper." "Furthermore[,] the environment of HALMA was not very succinctly described.
>
> As R1 pointed out, choosing which contents to remain in the main text is challenging for writing this paper, and we have to make a difficult choice. One way is like what R1 suggested; we could reserve the majority of the paper to describe the new environment's details. However, we argue this detail-oriented style would obscure the contributions of our paper.
>
> An alternative approach, which we believe is much better and eventually adopted, is to present the high-level ideas and well-motivate our work in the main text. The technical implementations are left to the appendix as a technical report for readers interested in details.
>
> Our intent to introduce HALMA is to propose a new research program instead of merely serving as an additional benchmark on existing problems. What we hope to deliver to readers in limited space are
> * what the problem is;
> * why we study it;
> * where we are now;
> * what we provide to facilitate this research; and
> * how we evaluate future progress.
>
> We hope readers can have both an intuitive understanding of the problem and an awareness of this research program's conceptual novelty. It would be better if they are thus intrigued to join or inspired to think about something more fundamental.
>
> Specifically, the conceptual novelties of our work, which also cover most contributions of our work, based on our own assessment, may include the following aspects:
>
> 1. An explicit definition of a concept space that is both minimum and complete, which indeed facilitates agents' rapid problem solving.
>
> By "minimum," we mean this concept space only involves probably the simplest structure in mathematics, natural numbers with elementary arithmetic. By "complete," we mean this concept space provides a comprehensive account of a spatial-temporal-causal (STC) structure, containing both "semantics" and "affordances." We firmly believe that only by having a clear definition of a concept space as the ground-truth can we measure how well artificial agents have learned about it, especially when the agent is designed to be "model-free" and "emerging." This very idea itself is novel to our community.
>
> 2. A careful design of the problem space such that the embodied agents' understanding of the concept space can be reflected in how they explore and plan.
>
> Even though similar concept spaces were introduced (e.g., CLEVR [1], PGM [2]), none explicitly accounts for the process of "rapid problem-solving." That is, the algorithmic generalization based on concept spaces, introduced in Sec. 2, has barely been measured systematically and has to be measured following a similar paradigm as proposed in this work.
>
> 3. A systematic construal with a suite of tests on agents' capability in learning towards and reasoning over the concept space.
>
> By categorizing our tests into "semantics," "affordance," and "analogy," we systematically dissect the measure of different aspects in agents' learning. This is another blessing of the complete concept space in HALMA, which, as far as we know, has not been discussed in prior arts.
>
> 4. A dynamically generated benchmark with tailored test problems to informatively measure learning agents' capability in explicit out-of-distribution generalization.
>
> Given the STC structure of the concept space, dynamic test problem generation should be one correct (and might be the only) direction if future works would follow the paradigm proposed here. This is primarily because of the difficulties to precisely constrain embodied agents to be exposed to certain configurations in the concept space; checking what they have not experienced is much easier.
>
> We agree with R1 that the algorithm we use to generate test problems itself is a technical contribution. Nevertheless, we reckoned that this specific algorithm's impact is not as significant as the idea that leads to its invention. After all, due to its strong dependency on the concept space, its technical **details** are highly unlikely to be reused by future works inspired by ours; they are likely to design new concept spaces and algorithms to spur new progress. It is the **idea** itself that can be generalized to future benchmarks.

---

> > ### Author Response · Authors · 2020-11-18
> > **Response to R1 (2/2)**
> >
> > > I was curious to what effect many of the research findings depended on the arbitrary size of the training set; and would have been interested to see some generalisation metrics reported as a function of training set size."
> >
> > We did report what you request in Appendix G.1. Specifically, we did an ablation study in terms of the volume of training mazes, covering four setups: 100, 300, 500, and 1000 mazes. Fig. S14 illustrates the main results.
> >
> > *Specifically, there is a significant promotion for the metric of goal reaching rate $\mathbf{\rho_g}$ in the challenging dynamic testing (from $30\text{-}60\%$ to $80\%$). More interestingly, starting from 300 training mazes, the distinction between different inductive biases vanishes. While the efficiency ratio $\mathbf{\rho_p}$ could also benefit from increased exposure, it reaches only around $50\%$ at best. As for the ratio of invalid moves $\mathbf{\rho_a}$, even though it reaches around $10\%$ in random split for stateless agent when trained with 1000 mazes, no clear trend can be detected in dynamic testing overall, which may suggest agents' limitation in understanding affordance with the temporal grammar or under the long-tail distribution of digits.*
> >
> > In sum, the increase in training volume does not significantly boost the metric of "invalid action" or "planning efficiency." These two metrics, however, explicitly account for agents' understanding of affordance. We believe this result validates our argument that new designs are needed for embodied agents and that HALMA does make a timely contribution to the community.
> >
> > > I would have also liked to see a human baseline and then to have had human-normalised scores.
> >
> > We thank your suggestions. However, the design of human baselines that **precisely** mimic HALMA is not trivial and deserves a dedicated paper reviewed by experts in cognitive science, which we reserve for our future work. Unfortunately, our subject pool is closed due to COVID-19, so we may not report the **naive** human study results during this review process. Nevertheless, we actively seek collaboration with developmental physiologists to develop a more scientific plan for a fair comparison between machines and humans. Below, we provide established evidences that endorse our claims.
> >
> > Even though the capability of doing 1-digit arithmetic may not be developed in the earliest 1-3 months, it is indeed developed relatively early in childhood, highly possible with the assistance of language [3]. This brings some intellectual challenges to the design of the human study. As written by Chollet in the manuscript raised by R4, "to make sure that humans test-takers do not bring further priors to the test, the test tasks should not rely on any acquired human knowledge (i.e., any knowledge beyond innate prior knowledge). For instance, they should not rely on language or learned symbols (e.g., arrows)." We can indeed design a naive human study that reduces this prior by replacing MNIST digits with Omniglot characters. However, since most adults are literate and can do arithmetic or counting quite robustly, the result from such human study is unfair to agents who are not assumed with language in HALMA.
> >
> > [1] Johnson et al., CLEVR: A Diagnostic Dataset for Compositional Language and Elementary Visual Reasoning, CVPR 2017 Oral
> >
> > [2] Barrett et al., Measuring abstract reasoning in neural networks, ICML 2018 Oral
> >
> > [3] Dehaene, The Number Sense: How the mind creates mathematics, 2011

---

### Official Review · AnonReviewer2 · 2020-10-30
**Useful benchmark that evaluates the task-solving capabilities of agents using the notion of affordances**

**Rating:** 6
**Confidence:** 3

**Review:**

---Post rebuttal---

Thank you for the detailed response. Overall, I think the proposed work provides a valuable benchmark for testing generalization ability of RL agents. However, I agree with R3 regarding the writing being dense/difficult to follow. I keep my rating unchanged (Weak Accept).

----

This work proposes a benchmark for evaluating the task-solving capabilities of agents on three levels: perceptual, conceptual, and algorithmic. The tasks are procedurally-generated contextual 2D gridworld environments.

I believe there is a lack of RL benchmarks on evaluating the agent’s understanding of object affordances, so this is a useful benchmark for the RL community.

Clarity: The paper is well-written and well-motivated.

Suggestions:

1. The paper provides empirical evaluations of TD3 with various encoder/decoder architectures. However, there does not seem to be evaluations of model-based/planning methods, despite the task requiring planning & reasoning. I think the comparison of model-free vs. model-based on this benchmark would be valuable.

2. For future work, I think it would be valuable to add continuous control to the benchmark tasks for more “humanlike abstraction learning”. For example, use locomotion actions instead of gridworld actions; or a robotic arm learning to play a logical puzzle game.

3. The related works section (Appendix H) can also add prior work on visual semantic navigation, which connects visual and semantic understanding with control:
[1] VIsual Semantic Navigation using Scene Priors https://arxiv.org/pdf/1810.06543.pdf
[2] Embodied Multimodal Multitask Learning https://arxiv.org/pdf/1902.01385.pdf

---

> ### Author Response · Authors · 2020-11-18
> **Response to R2**
>
> We thank R2 for acknowledging our "paper is well-written and well-motivated" and sharing the same concern that "there is a lack of RL benchmarks on evaluating the agent's understanding of object affordances," one of our main contributions. We hope our detailed responses below would resolve the remaining questions you have and therefore increase the rating of our work.
>
> > The paper provides empirical evaluations of TD3 with various encoder/decoder architectures. However, there does not seem to be evaluations of model-based/planning methods, despite the task requiring planning & reasoning. I think the comparison of model-free vs. model-based on this benchmark would be valuable.
>
> This is a misunderstanding of our work. As mentioned throughout the main text oft-repeatedly, the thesis argument of this paper is model-free agents to date still lack proper inductive biases to learn a complete humanlike abstraction as defined in HALMA. Note that [3][4] have reported that model-free agents achieved certain generalization that is behaviorally equivalent to model-based/planning agents. In contrast, the results revealed by our experiments offer the first substantial piece of evidence that refutes their findings, rendering their arguments speculative.
>
> If we were to take one step back, evaluations on model-based agents could complement the current benchmark. However, in theory, a naive model-based RL agent would still struggle in HALMA. HALMA is a partially observable task in which the layout of walls and crossings in each problem are randomly different. Hence, a complete world model learned in one problem is of limited usage to another novel problem, unless the agent could master the underlying concept space to afford generalizations across problems. Furthermore, HALMA's option space is combinatorially large, thus preventing a naive adoption of model-based RL agents.
>
> Let us elaborate on the above explanation with a concrete example. Consider the example trial in Appendix A.3; see Fig. S4: In Step 1 (Fig. S4(a)), an oracle agent reads from the visual panel---there is one empty block in its up direction. After it moves right for 2 and reaches the state depicted by Fig S4(b), it finds there is still one empty block on its up direction. Will this always happen for this action? Undoubtedly no. This only happens because the maze is randomly generated to be like this; it is not caused by the action the agent takes. And it would be improbable to happen in other scenarios. If the agent had moved right for 1, it would see no empty block in its up direction. Such a transition is coined as "exogenous" in classical AI, meaning something relating to or developing from external factors. Crucially, exogenous factors would challenge classical planning if the agent failed to realize they are exogenous. And these factors would introduce substantial noises to the learning of naive model-based agents, hindering their success in the setup of **rapid** problem solving.
>
> In sum, a success in HALMA would require strong capability in understanding the concept space and forming logical and deliberative strategies, just as we humans do in System 2 [5]. The expected conceptual generalization and algorithmic generalization would even spur the study of model-based learning methods.
>
> Still, we would like to thank R2 for constructive suggestions. We do notice that model-based RL agents were barely evaluated in tasks with rich concept spaces. We will include this component in our future work.
>
> > For future work, I think it would be valuable to add continuous control to the benchmark tasks for more "humanlike abstraction learning". For example, use locomotion actions instead of gridworld actions; or a robotic arm learning to play a logical puzzle game.
>
> Excellent suggestion. A continuous action space will definitely make this problem more challenging, and is certainly what we intend to investigate as future work. However, by isolating continuous actions, the HALMA environment becomes much more focused on the core challenge we hope to study---the role of affordance in forming the concept space.
>
> > The related works section (Appendix H) can also add prior work on visual semantic navigation, which connects visual and semantic understanding with control: VIsual Semantic Navigation using Scene Priors, Embodied Multimodal Multitask Learning."
>
> Thanks for bringing these works to us. We have added a brief introduction in Related Work.
>
> [3] Wang et al., Prefrontal cortex as a meta-reinforcement learning system, Nature Neuroscience 2018
>
> [4] Guez et al., An investigation of model-free planning, ICML 2019 Oral
>
> [5] Kahneman, Thinking, fast and slow, 2011

---

### Official Review · AnonReviewer3 · 2020-11-03
**Dense, but well motivated**

**Rating:** 7
**Confidence:** 2

**Review:**


Summary
---

Children generalize to new sights that combine known perceptual elements.
Children generalize to new instances of known abstract concepts like order and number.
Children know what actions they can take in new scenarios, because those actions have been available in similar contexts.
Machines should be able to generalize in the same ways.

This paper proposes a new environment, HAMLA, where machines are tested on their
ability to generalize in all of these ways.
Agents navigate through a maze to a goal location using only carefully designed
signals extracted from carefully constructed mazes.
They must learn perception (MNIST digits, color, location), abstract concepts
(number, order), affordances (move up/down/left/right as available), and
efficient exploration strategies at once, like humans seem to be able to.

Evaluation is dynamic, estimating what an agent already knows then generating
new test instances that test the model on something slightly outside what it knows.
There is no static test set or test environment.

TD3 is used to train various agents based on different NN architectures
that incorporate more or less structure.
Agents often fail to navigate to the goal and do so efficiently in scenarios basically similar to those it was successful at.
Failure happens more often as the generalization gap becomes larger and when the agents must also learn perception in addition to concepts and exploration.


Strengths
---

The motivation is ambitious, interesting, and relevant. It makes sense. It pulls strongly from cognitive science. Explicitly attacking multiple specific modes of generalization at once is an interesting direction.

This could establish a new baseline task that tests multiple kinds of generalization in a toy manner.

The dynamically generated evaluation strategy is new and interesting. It may make it harder to compare performance across models, but clearer about how well a single model is actually performing. That could inspire a significant shift in evaluation methodology.

The paper is highly sylized and polished.


Weaknesses
---

1) The paper is hard to understand in just 8 pages.

The writing is so dense and so many details are left to the 20 page appendix that it is hard to understand the approach or experiments at more than a very high level by reading just the main 8 pages of the paper.

The main example here is the notation. Much of it is non-standard. It is also used frequently and most definitions are left for the appendix.


2) Novelty relative to some related work isn't clear.

How does this compare to point-goal nav? [1] I think the test procedure and available senses make it different, but it's still fundamentally navigating to a goal location. Will scaling training help solve this problem as evaluated by the proposed metric?

[1]: Wijmans, Erik et al. “DD-PPO: Learning Near-Perfect PointGoal Navigators from 2.5 Billion Frames.” ICLR (2020).


3) Impact may be limited by difficulty. Tackling this whole problem may be too difficult right now. All approaches completely fail to solve the full problem including vision, as indicated by the bottom right of table 1, which is filled with 0s. If progress is limited to the symbolic setting then the problem is significantly less interesting.


Preliminary Evaluation
---

At the moment the paper is not very clear. That makes it hard to evaluate the quality of the experiments. The quality and novelty of the motivation is high, being fairly novel and interesting. Its significance is highly uncertain because of the paper's clarity and potential difficulty. The paper might be significant as either 1) a central reference for applying some cognitive science concepts to AI, 2) a benchmark that spurs new agent designs, or 3) inspiration for designing new evaluation metrics.

My main uncertainty in this evaluation is because I haven't understood the paper in its full 30 pages of depth.

---

> ### Author Response · Authors · 2020-11-18
> **Response to R3 (1/3)**
>
> We sincerely thank R3 for acknowledging our work "ambitious, interesting, and relevant," "[e]xplicitly attacking multiple specific modes of generalization at once is an interesting direction," "[t]he dynamically generated evaluation strategy is new and interesting," and our design "could inspire a significant shift in evaluation methodology." Below, we provide point-to-point responses to address all the questions that hopefully would increase your rating of our work.
>
> > There is no static test set or test environment.
>
> Actually, there is a static test, which we refer to as Random Split in the manuscript, although there is indeed no static test for explicit generalization.
>
> > The main example here is the notation. Much of it is non-standard. It is also used frequently and most definitions are left for the appendix.
>
> We respectfully disagree. We adopt **standard** notations from reinforcement learning, classical AI, and mathematics. Some notations may seem non-standard to colleagues in the machine learning community as the denoted notions are unfamiliar to this community. For such notations, we have cited original, classic, and well-known sources throughout the manuscript. Due to our work's interdisciplinary and original nature, we firmly believe that understanding standard notations in other fields is what we have to overcome if we were to study beyond existing paradigms. We tried our best to make all notations and notions formal to prevent ambiguity; please kindly advise if any specific ones are confusing.
>
> > How does this compare to point-goal nav? I think the test procedure and available senses make it different, but it's still fundamentally navigating to a goal location. Will scaling training help solve this problem as evaluated by the proposed metric?"
>
> This is a misconception; HALMA is fundamentally different from existing 3D visual navigation tasks. And it is primarily the design of the temporal grammar for the option space that buttresses this distinction. Recall the anecdote of Ada playing Halma in the Introduction; the most exciting aspect of Halma, or more specifically Super Halma, is that you can either move a pawn for one hole, catapult for more holes, or recursively combine them for even fancier moves, depending on which ones are affordable given the context. These moves are only counted as one time-step. In contrast, since Habitat's action space only contains 5 actions, those fancy "jumpy moves" have to be counted as multiple time-steps. Obviously, the optimal solution in HALMA would have significantly fewer time-steps than the optimal in Habitat.
>
> We also provided results in the ablation study (Appendix G.2) to support this claim. We do find that if the max_opt_len=1 (an action space of size 13 that is close to the size of 5), LSTM agents, Transformer agents, and LSTM+Transformer agents all perform exceptionally well in our dynamic tests, consistent with the result in DD-PPO [1]. However, all metrics degrade with the max_opt_length increases, indicating the incompetence in understanding the concept space. In other words, the problem of affordance is only interesting if the action/option space is large enough. For example, when the agent sees a 4 in the visual panel, it ought to understand 1, 2, 1+2, 1+1+2,... are all affordable moves, and there are also moves such as 2+3+1, 2+2+2 that are not affordable. This is where combinatorial generalization excels.
>
> It is also worth noticing that the concept space in Habitat might be too simple. After all, it is not a visual semantic navigation task as raised by R2. The only useful concepts may be "direction with greater depth are walkable," "walking towards goal." The conceptual generalization that we characterize cannot be evaluated in Habitat.

---

> > ### Author Response · Authors · 2020-11-18
> > **Response to R3 (2/3)**
> >
> > > Impact may be limited by difficulty. Tackling this whole problem may be too difficult right now. All approaches completely fail to solve the full problem including vision, as indicated by the bottom right of table 1, which is filled with 0s. If progress is limited to the symbolic setting then the problem is significantly less interesting."
> >
> > Without getting too philosophical, **we firmly believe that the proposed HALMA makes a timely contribution**, detailed below.
> >
> > Even though HALMA provides a holistic evaluation of perceptual, conceptual, and algorithmic generalization, **it does not mean we have to tackle the challenge as a whole**. In fact, Table 1 is a good example of dissecting this grand problem into actionable sub-problems, the progress on which can hopefully bring improvement to the problem as a whole.
> >
> >
> > First, **one can always start with some sub-problems first in the symbolic setup**. Let us take the perspective of the evaluation metrics. The primary problem that is meaningful in the symbolic setup would be the very idea of **learning affordance**. After all, an oracle visual module can only resolve the problem of **semantics**. This argument has been further backed up in our ablation study in Appendix G.1: Adding training data by 10 times does not bring significant improvement to the metric of valid actions, indicating that agents do not understand the combinatorially large concept space, in particular the temporal grammar. Hence, even with perfect perception, proper inductive biases are still in demand. Similarly, another sub-problem that could be studied stand-alone is the improvement of planning efficiency. Moreover, we can modularize our study according to the test category. In the Analogy Test, agents do not benefit from their inductive biases, on which new inductive biases may also in need.
> >
> > Second, **the problem in the visual setup may not be as daunting as it seems**. When we designed the one-hot symbolic input, we did a simple controlled experiment to compare with continuous symbolic input. That is, instead of using a one-hot encoding of each category, we use one entry ranging from 0 to 9. Surprisingly, the performance of the agent degrades significantly even in terms of the data efficiency of training. The point is, most works we cited when introducing perceptual generalization aimed to improve the efficacy in downstream cognitive activities such as thinking and reasoning; however, there does not exist a benchmark to systematically test their models toward this claim, not until HALMA. A natural question arises: Is it possible that a simple revision in the format of the latent variable can significantly improve the visual problem in HALMA? After all, what we discovered in our experiment is only about continuous latent variables. What if the latent variable learned in unsupervised training is discrete [2]? We believe our discovery can definitely evoke some fundamental discussions in the community of unsupervised disentanglement. We also believe new progress is not far away because researchers have already realized the trade-off in the formatting of latent variables [3].
> >
> > Third, **other customizations can be designed with HALMA to exclude problems of less interest**. For example, if the perceptual generalization, particularly generalizing to unseen digits, is not regarded as important, but still the dimensionality of visual problems is of interest, test problems can be generated with the training image set. The point is, although we provide a complete suite in HALMA, researchers do not need to use all the building blocks in this suite for actionable research. This is precisely the human-like conceptual generalization characterized in this manuscript, isn't it? We will add this discussion of modularizing difficulty into the main text in the final revision.
> >
> > [2] Vahdat et al, DVAE++: Discrete Variational Autoencoders with Overlapping Transformations, ICML 2018
> >
> > [3] Jiang and Ahn, Generative Neurosymbolic Machines, NeurIPS 2020 Spotlight

---

> > > ### Author Response · Authors · 2020-11-18
> > > **Response to R3 (3/3)**
> > >
> > > Further, for researchers who wish to solve the problem as a whole, we are also thrilled to find some parallel progress reported in this venue [4][5]. In [4], agents learn to ground compositional nature language sentences with 3D navigational visuomotor experience and achieve certain systematic generalization in language understanding. The agent also achieves above-chance probability in generalizing over a category, from a seen car to an unseen car (Fig. 4), which is a variant of our Semantic Test. In [5], agents memorize the whole map for rapid problem solving, a task formulated similarly to ours. They demonstrated a close-to-oracle performance in rapid visual navigation (Fig. 2 and 3) that does not involve a concept space. **Both papers have received very positive reviews.** However, [4] provides language as supervision and does not involve Affordance Test. [5] only needs to memorize the whole map, rather than understanding the concept space. If the authors of these works keep moving forward, they would probably need a challenge similar to HALMA. Actually, it is literally written in the Future Work of [5] that "temporally abstract planning or, 'jumpy' planning" and "open-ended challenge like Frostbite... that adopts generative grammar" may be their future directions. This is exactly what we have in HALMA.
> > >
> > >
> > > [4] Anonymous, Grounded Language Learning Fast and Slow. https://openreview.net/forum?id=wpSWuz_hyqA
> > >
> > > [5] Anonymous, Rapid Task-Solving in Novel Environments. https://openreview.net/forum?id=F-mvpFpn_0q

---

### Public Comment · ~Anirudh_Goyal1 · 2020-11-10
**Really Nicely written**

Dear. Authors,

I hope everyone is doing well in such times.

I want to mention that your work is exceptionally well written. May be too well written  :-)

I really like the idea of making analogies by leveraging different affordances of  objects.

Thanks for your time, in reading my message.

---

> ### Author Response · Authors · 2020-11-18
> **Thank you**
>
> Thank you very much! We do believe that analogy with affordance is an exciting problem. An exemplar phenomenon that reflects this kind of reasoning is: What makes a chair a chair? We do not merely recognize chairs from their appearance or geometry; rather, anything could be repurposed as a chair so long as it can afford a (comfortable) sit. A computational framework for such task-oriented abstraction is definitely in need for humanlike agents. However, this problem is only interesting when an object affords multiple actions and one action is affordable for multiple objects. We believe our benchmark is a long overdue one for our community.

---

### Author Response · Authors · 2020-11-18
**Manuscript updated**

Dear reviewers,

Thank you for your helpful comments. Per your request, we have revised the manuscript in:

* Moving the Related Work to the main text and adding discussion on the relation with Habitat and Visual Semantic Navigation and Control;
* Fixing some typos.

---

### Decision · Program_Chairs · 2021-01-07
**Final Decision**

**Decision:**

Reject

**Comment:**

This paper proposes a new task domain for learning-based AI agents, HALMA, a game that is designed to bring together multiple areas of research in AI. Perception, in the form of recognition of MNIST digits, learning mathematics - in the form of arithmetic operations on the natural numbers, and navigation and planning. When combined into a game, these elements are argued to require various important properties of human cognition, such as abstraction, analogy and affordance.

I commend the ambitious goals of this work, and its multidisciplinary motivations. I believe the benchmark can indeed eventually be an important challenge for the community. I think that the dynamic testing aspect is particularly interesting, where the environment produces trials designed to go beyond the agent's experience to that point. However, having considered the views of the reviewers and read the (main body) paper entirely myself, I unfortunately cannot recommend acceptance in its current form.

The main reason for the decision is simply that it is prohibitively challenging for me to grasp exactly how the game actually works after a thorough reading and considerable thought. The authors spend two pages motivating the approach with (arguably excessively grandiose) allusions to Marr's levels of analysis, Gibson's affordances, Holyoak's analogy and various other famous works from the history of AI and philosophy of mind; as well as to the board game HALMA. But as a reader I can't myself start to make any of these connections because the game proposed by the authors has not been explained to me! It is finally introduced on the fourth page - with reference to Figure 2 which is too small to consult and very hard to interpret. After consulting the appendix (where the idea is a bit clearer) I was able to decipher the way in which the numbers related to the maze itself, but was (and am still) unclear on the actions available to the agent. These are explained as follows:

""The direction set is t , , , u. The primitive action set, in terms of the numberof moves, is t , , , u; this design of primitive numbers with a maximum of three aligns withthe doctrine of core knowledge in developmental psychology (Feigenson & Carey, 2003; Dehaene,2011). If an option is selected, consecutive hops as in Halma are simulated; all observations fromintermediate states will be skipped, and only the observation of the final state is provided. A movewould fail if a wall stops the agent, leaving the agent’s position unchanged; failure moves bringpenalties to the agent. The agent would receive a positive reward when reaching the goal""

From reading this I am left with the following questions:
- Are directions primitive actions?
- How can a primitive action also be a number of moves?
- What does it mean to select an option?
- Can I select an option and an action at the same timestep?

Most importantly, I still don't really know how the game works.

This example is intended to illustrate the difficulty faced by readers of this paper in general.

I note that the reviewers awarded this work scores that place it on the borderline for acceptance, but with consistently low confidence. On consulting with the reviewers it is clear that this is not because they lack expertise but because they too did not understand the full details of how this domain/task works. This is also clear from the lack of detail in their reviews; only reviewer 3 engaged with any of the details of the task itself.

To summarise, I think there is potentially a very interesting and important contribution in this dataset. However, the work will only have impact in the community if it can be understood and adopted after a single read of the paper. I therefore recommend that the authors resubmit this work to a different venue taking account of the following:

- Explain how the game works *then* connect it to the literature on human learning *not* vice versa (from the concrete to the abstract)
- Be very concrete, perhaps guide the reader through a single particular episode explaining the observations available to the agent and the options open to it at each important point-
 Get to the point of your contribution. Tenuous connections to cogsci etc can go in the discussion
- Make all diagrams and illustrations extremely simple to interpret and large enough to easily read
- Avoid use of subjective adjectives, and particularly describing one's own contributions as "ingenious solutions" and "impeccable"
- Avoid rhetorical flourishes and latin
- Submission to a journal may allow the authors greater space to draw the desired connections to disparate fields without compromising on readability or exposition of their methods

---

> ### Author Response · Authors · 2021-02-04
> **Thank you for your meta review**
>
> Dear ACs,
>
> Thank you very much for your detailed meta-review. We understand our following response will not reverse the final result. Nevertheless, we hope our explanations below can help both you and other readers better understand our paper.
>
> > After consulting the appendix (where the idea is a bit clearer) I was able to decipher the way in which the numbers related to the maze itself, but was (and am still) unclear on the actions available to the agent. These are explained as follows:
>
> > ""The direction set is t , , , u. The primitive action set, in terms of the number of moves, is t , , , u; this design of primitive numbers with a maximum of three aligns with the doctrine of core knowledge in developmental psychology (Feigenson & Carey, 2003; Dehaene,2011). If an option is selected, consecutive hops as in Halma are simulated; all observations from intermediate states will be skipped, and only the observation of the final state is provided. A move would fail if a wall stops the agent, leaving the agent’s position unchanged; failure moves bring penalties to the agent. The agent would receive a positive reward when reaching the goal""
>
> > From reading this I am left with the following questions:
>
> > * Are directions primitive actions?
> > * How can a primitive action also be a number of moves?
> > * What does it mean to select an option?
> > * Can I select an option and an action at the same timestep?
>
> In the very sentence prior to your quote, we already provided the answers to your above questions: ''When making a decision, the agent needs to **first** select a direction and **then** select **either** a primitive action **or** an option composed by a sequence of primitive actions (Sutton et al., 1999) with maximum length max_opt_len.'' Could you please kindly advise if there is any ambiguity in this sentence? Thank you very much!